# FairCoT: Enhancing Fairness in Diffusion Models via Chain of Thought Reasoning of Multimodal Language Models

## Abstract

In the domain of text-to-image generative models, biases inherent in training datasets often propagate into generated content, posing significant ethical challenges, particularly in socially sensitive contexts. We introduce FairCoT, a novel framework that enhances fairness in diffusion models through Chain-of-Thought (CoT) reasoning within multimodal generative large language models (LLMs). FairCoT employs iterative CoT refinement and attire-based attribute prediction to systematically mitigate biases, ensuring diverse and equitable representation in generated images. By integrating iterative reasoning processes, FairCoT addresses the limitations of zero-shot CoT in sensitive scenarios, balancing creativity with ethical responsibility. Experimental evaluations across multiple models, including DALL-E and various Stable Diffusion variants, demonstrate that Fair-CoT significantly improves fairness and diversity metrics without compromising image quality or relevance. Our approach advances ethical AI practices in generative modeling, promoting socially responsible content generation and setting new standards for fairness in AI-generated imagery.

## 1 Introduction

The advent of text-to-image generative models has revolutionized artificial intelligence, enabling the synthesis of high-fidelity images from textual descriptions. These advances have unlocked new possibilities in creative expression, design, and accessibility. However, these models often inherit and amplify biases present in their training datasets, leading to ethical concerns—especially when generating content related to sensitive social issues. Addressing these biases is critical to ensure that AI systems are fair, inclusive, and socially responsible.

Existing approaches to mitigate bias in text-to-image models primarily focus on prompt engineering or adjusting model parameters. Handcrafted prompting methods (Bianchi et al., 2023; Bansal et al., 2022) rely heavily on human annotations, which are inherently subjective and can introduce inconsistencies (Sun et al., 2023). Moreover, these methods can be costly and sub-optimal due to the extensive manual effort required. On the other hand, debiasing techniques involving parameter fine-tuning (Gandikota et al., 2024; Shen et al., 2023) or modifying text embeddings (Chuang et al., 2023) are computationally intensive and often limited to open-source diffusion models. These methods can inadvertently affect model alignment and are typically restricted to specific types of biases addressed during fine-tuning (Sun et al., 2023).

To address these challenges, we introduce FairCoT, a novel method that leverages Chain-of-Thought (CoT) reasoning within multimodal large language models (MLLMs) to refine the generative process of text-to-image models. FairCoT enables models to identify and mitigate biases in their outputs by generating and refining fairness-aware reasoning paths. It operates by integrating iterative reasoning refinement and controlled evaluation to guide generation toward more diverse and equitable outputs. Leveraging MLLMs' reasoning capabilities, FairCoT assesses potential biases and adjusts the generation accordingly. During the CoT generation phase, it generates reasoning steps that consider fairness constraints related to sensitive attributes (e.g., gender, race, age, and religion). Feedback from the controlled evaluations refines the reasoning steps in subsequent iterations, allowing dynamic adjustment of outputs based on fairness considerations. At inference, the MLLM adapts the

most relevant Chain-of-Thought to the new task. It then generates a set of text prompts inspired by this adapted reasoning to guide the diffusion model in producing fair outputs.

This approach addresses limitations of zero-shot CoT reasoning in sensitive contexts by providing explicit, iterative fairness guidance. Unlike prompt engineering or fine-tuning methods, FairCoT operates at the reasoning level, making it applicable to both open and closed-source models without parameter updates. By leveraging MLLMs' reasoning capabilities, FairCoT directly influences the generative process to align with ethical objectives, ensuring outputs are high-quality and socially responsible. Our contributions are summarized as follows:

1. **FairCoT Framework:** We introduce FairCoT, a simple and effective method that applies Chain-of-Thought (CoT) reasoning for bias reduction in text-to-image (T2I) models. It avoids retraining, works for both open and closed-source models, and preserves alignment.

2. **Multi-Bias Generalization:** FairCoT addresses multiple bias types simultaneously, including sensitive attributes and objects, and generalizes well across domains.

3. **Improved Attribute Prediction:** We present an attire-based method using CLIP for improving the detection of sensitive attributes like religion, making us the first to tackle religious bias in T2I models.

4. **Iterative Bootstrapping with MLLMs:** We apply an iterative bootstrapping process using MLLMs and CoT for debiasing, the first use reasoning for T2I models and fairness effectively.

FairCoT represents a significant advance in the pursuit of equitable and responsible AI. It is applicable to both open and closed-source text-to-image models and can address multiple bias types simultaneously. By integrating this method with LLM-powered diffusion models like DALL-E, we demonstrate its robustness and versatility. Furthermore, FairCoT can be applied upon request, helping to avoid overrepresentation or underrepresentation of minorities in specific contexts.

The remainder of this paper is structured as follows. In Section 2, we review related work on bias mitigation in generative models. Section 3 details the technical architecture of FairCoT, including the iterative reasoning refinement process and the integration of attire-based attribute prediction. Section 4 presents the experimental results with comprehensive analysis, and Section 5 discusses the broader implications of our work and potential future directions.

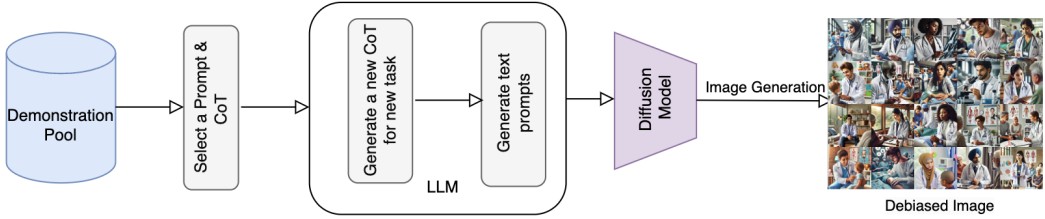

Figure 1: FairCoT Inference, where an MLLM generates text prompts from Chain of thought to produce fair images in Diffusion models

## 2 RELATED WORK

This section discusses relevant work on bias in vision-language models, the impact of dataset biases, language-driven bias mitigation, advances in large language models, and attribute prediction.

**Bias in Vision-Language Models** Bias in vision-language models is a significant concern, as these models often inherit and amplify societal biases present in training data. Hall et al. (2023) investigated gender bias in zero-shot vision-language models, revealing performance disparities and calibration issues based on perceived gender in images, highlighting how language influences both expanding and biasing vision tasks. Similarly, Kim et al. (2023) proposed a bias-to-text (B2T) method using pre-trained vision-language models to detect visual biases by generating captions

for misclassified images and identifying bias-indicative keywords, effectively pinpointing biases without human annotation.

Shrestha et al. (2024) introduced FairRAG, focusing on mitigating biases in human generation tasks by incorporating fair retrieval methods. Their approach adjusts retrieval processes to include diverse and representative examples, reducing biases in generated content. On the other hand, Shen et al. (2023) proposed fine-tuning text-to-image diffusion models for fairness using a single textual inversion token, demonstrating that targeted fine-tuning can mitigate demographic biases, although it may be computationally intensive and limited to specific models. Furthermore, He et al. (2024) presented the Iterative Distribution Alignment (IDA) method, addressing social biases in T2I diffusion models by guiding the diffusion process away from biased outputs through iterative weight updates based on Kullback-Leibler (KL) divergence. However, these methods often require extensive computational resources or are limited to certain models, highlighting the need for a more efficient and generalizable approach to bias mitigation in T2I models.

**Impact of Dataset Bias**    Dataset biases significantly impact vision-language models. Garcia et al. (2023) highlighted widespread societal biases in datasets across tasks like image captioning, text-image classification, and text-to-image generation, emphasizing the need for careful data curation. Additionally, Birhane et al. (2023) showed that larger datasets can amplify biases. Janghorbani & de Melo (2023) developed the MMBias benchmark for assessing bias, contributing to debiasing methods using additive residuals. Moreover, the PATA dataset introduced by Seth et al. (2023) provided insights into the effect of data on bias, underscoring the necessity of addressing imbalances in training data.

**Language-Driven Approaches in Bias Mitigation**    Language-driven strategies have been proposed for bias mitigation. Chuang et al. (2023) used text embedding projection for debiasing vision-language discriminative and generative models. On the other hand, Smith et al. (2023) proposed dataset debiasing by enhancing datasets with synthetic, gender-balanced sets. In text-to-image models, Bianchi et al. (2023) suggested adding gender terms to prompts to balance gender representation in images. Similarly, Bansal et al. (2022) advocated adding ethical statements to prompts to directly encourage fairness.

**Advances in Large Language Models**    Recent advances in LLMs have enhanced model reasoning and debiasing capabilities. Sun et al. (2023) introduced iterative bootstrapping in Chain-of-Thought (CoT) prompting to improve problem-solving capabilities in models like ChatGPT. Zelikman et al. (2022) proposed Self-Taught Reasoner (STaR), enabling models to self-supervise and refine reasoning steps without additional labeled data. Furthermore, Lyu et al. (2023) focused on improving the faithfulness of reasoning steps in LLMs, addressing hallucination, and enhancing reliability in sensitive contexts. However, Shaikh et al. (2022) highlighted risks of zero-shot CoT reasoning in socially sensitive domains, showing that models may generate biased or harmful content without proper guidance, underscoring the need for mechanisms to minimize biased outputs.

**Attribute Prediction**    Attribute prediction is crucial for assessing demographic representations in generated content. CLIP (Radford et al., 2021) has been widely used for zero-shot image classification and attribute prediction due to its ability to learn visual concepts from natural language supervision. Radford et al. (2021) found high accuracy (96%) for gender classification across all races using CLIP. They averaged around 93% for racial classification and approximately 63% for age classification. In addition, Shen et al. (2023) adopted the eight race categories from the FairFace dataset but found classifiers struggled to distinguish between certain categories. Therefore, to improve race attribute prediction, they consolidated them into four broader classes: WMELH (White, Middle Eastern, Latino Hispanic), Asian (East Asian, Southeast Asian), Black, and Indian. Han et al. (2017) presented a deep multi-task learning approach for heterogeneous face attribute estimation, emphasizing multi-task learning to improve attribute prediction accuracy and robustness across diverse populations.

Our work builds upon these studies by integrating Chain-of-Thought reasoning within multimodal generative LLMs to enhance fairness in diffusion models. By leveraging iterative reasoning and attire-based attribute prediction, we address limitations identified in previous works, contributing to ethical AI practices in text-to-image generation.

## 3 METHODS

In this section, we introduce FairCoT, a framework designed to enhance fairness in AI-generated imagery by integrating Multimodal Large Language Models (LLMs) and Contrastive Language-Image Pre-training (CLIP). We also propose an attire-based method for predicting religious attributes, aiming to improve CLIP's attribute prediction for challenging tasks, leading to accurate automatic image labeling. The framework operates in two primary phases: CoT generation and inference, each comprising several key components.

### 3.1 CHAIN OF THOUGHT GENERATION PHASE

#### 3.1.1 INITIAL IMAGE GENERATION

We begin by defining a set of professions $\mathcal{P} = \{p_1, p_2, \ldots, p_n\}$ that are commonly depicted in the working industry. These professions are chosen to cover a wide range of societal roles to ensure the generalizability of our approach, each belonging to one of the following areas: Healthcare and Medical, Legal and Business, Service and Hospitality, Security and Protection, Education and Information, Engineering and Technical, and Research and Analytics.

To represent diversity, we identify demographic attributes $\mathcal{D} = \{\text{gender}, \text{age}, \text{race}, \text{religion}\}$ that are essential. For the attribute of religion, we further specify a set of religious groups $\mathcal{R} = \{r_1, r_2, \ldots, r_k\}$. We utilize a Multimodal Large Language Model to generate prompts that specify a profession. Formally, for the given set of professions $\mathcal{P}$, the LLM produces a set of prompts $\mathcal{S} = \{s_1, s_2, \ldots, s_n\}$, where each prompt $s_i$ is initially "n photos of $p_i$".

The generated prompts $\mathcal{S}$ are then used to guide the text-to-image model to produce an initial set of images $\mathcal{I} = \{I_1, I_2, \ldots, I_m\}$. This stage is critical as it provides the raw data for subsequent bias evaluation and refinement processes (see Figure 3).

#### 3.1.2 ENHANCED ATTRIBUTE PREDICTION

We predict the attributes for the generated images using CLIP zero-shot classification. For attributes that CLIP does not accurately predict due to their complex nature, we propose an attire-based method to improve the prediction.

We input the set of religious groups $\mathcal{R}$ into the LLM, which generates a detailed list of attires associated with the corresponding religions. This step recognizes the diverse visual expressions of religious identities. The LLM outputs an attire list $\mathcal{A} = \{a_1, a_2, \ldots, a_l\}$, where each attire $a_j$ corresponds to a specific religious group (e.g., hijabs for Islam, turbans for Sikhism, kippahs for Judaism).

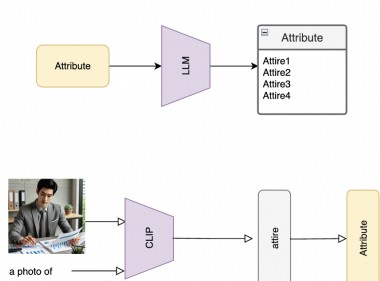

To analyze each generated image $I_i \in \mathcal{I}$, we employ CLIP to detect the most prominent attire $a_j \in \mathcal{A}$. This is achieved by computing similarity scores between the image embeddings and textual embeddings of the attires:

$$\text{Score}(I_i, a_j) = \cos\left(\phi_I(I_i), \phi_T(a_j)\right),$$

Figure 2: Improving CLIP attribute prediction

where $\phi_I$ and $\phi_T$ are the image and text embedding functions of CLIP, respectively. The attire with the highest score is selected and mapped to the corresponding religion, which will be the predicted attribute for that image.

#### 3.1.3 BIAS AND ALIGNMENT EVALUATION AND ITERATIVE BOOTSTRAPPING

To assess the uniformity of the attribute distribution across the images, we compute the normalized entropy $H'$ over the set of demographic attributes:

$$H' = -\left( \frac{1}{\log k} \sum_{j=1}^{k} p(a_j) \log p(a_j) \right),$$

where $p(a_j)$ is the empirical probability of attribute $a_j$ appearing in the image set $\mathcal{I}$. We also measure the alignment between the generated images and the expectations set by the initial prompts of the professions using the CLIP-Textual (CLIP-T) score:

$$\text{CLIP-T Score} = \frac{1}{m} \sum_{i=1}^{m} \cos\left( \phi_I(I_i), \phi_T(s_i) \right),$$

where $s_i$ is the prompt used to generate image $I_i$.

### 3.1.4 CoT DEBIASING

Starting with an initial Chain-of-Thought $\text{CoT}_0$ obtained through zero-shot debiasing (Kojima et al., 2022), we prompt the MLLM with $r_0$: "Think step by step before generating images while considering several races, genders, religions, and ages and treating people of these categories equally," which generates $\text{CoT}_0$ (see Figure 3).

Having the initial Chain-of-Thought $\text{CoT}_0$ as a baseline, the model iteratively enhances fairness. In each iteration $t$, the MLLM refines the CoT based on the bias assessments until the images reach convergence criteria. The iterative process is as follows:

1. **Bias Evaluation:** Compute $H'_t$ and CLIP-T Score at iteration $t$.
2. **CoT Refinement:** If $(H'_t > H'_{t-1})$ and $(\text{CLIP-T}_t > \tau \text{CLIP-T}_{t_0})$, update $\text{CoT}_t$ to address the identified biases; where $\tau < 1$ is the acceptable drop in alignment for compromising fairness.
3. **CoT Generation:** Generate $CoT_t$ using prompt $r_t$: "Can you think again? Consider generating images of different religions, races, ages, and genders".
4. **Image Production:** Produce new images $\mathcal{I}_t$ using the diffusion model guided by $\text{CoT}_t$.
5. **Iteration:** Increment $t$ and repeat.

The iterative process continues until the fairness criteria are satisfied:

$$H'_t \leq H'_{t-1} \quad \text{or} \quad \text{CLIP-T Score} \leq \tau \text{CLIP-T}_{t_0}.$$

Upon convergence, we document the detailed $\text{CoT}_t$ that describes the iterative reasoning and adjustments made to achieve the fairness standards.

## 3.2 INFERENCE PHASE

### 3.2.1 DEMONSTRATION POOL AND APPLICATION

The images $\mathcal{I}$ that meet the fairness standards, along with their corresponding Chains-of-Thought CoT, prompts $s$, and professions $p$, are archived in a demonstration pool $\mathcal{D}_{\text{pool}}$. This repository serves as a template for future tasks, ensuring that the learned fairness principles are replicated in new contexts.

### 3.2.2 INFERENCE

For a new task involving a different profession $p_{\text{new}}$, we select an appropriate CoT from $\mathcal{D}_{\text{pool}}$ based on the seven areas considered for professions. The MLLM adapts this CoT to generate a new $\text{CoT}_{\text{new}}$ tailored to $p_{\text{new}}$ (see Figure 1). We utilize the MLLM to generate prompts for the new profession inspired by the new $\text{CoT}_{\text{new}}$. The LLM produces a set of prompts $\mathcal{S} = \{s_1, s_2, \ldots, s_n\}$, where each prompt $s_i$ is crafted to encourage diversity in the subsequent image generation process.

The adapted prompts $\mathcal{S}_{\text{new}}$ guide the diffusion model to produce images $\mathcal{I}_{\text{new}}$ that adhere to the established fairness principles. This ensures that the new images maintain the diversity and fairness standards set by FairCoT.

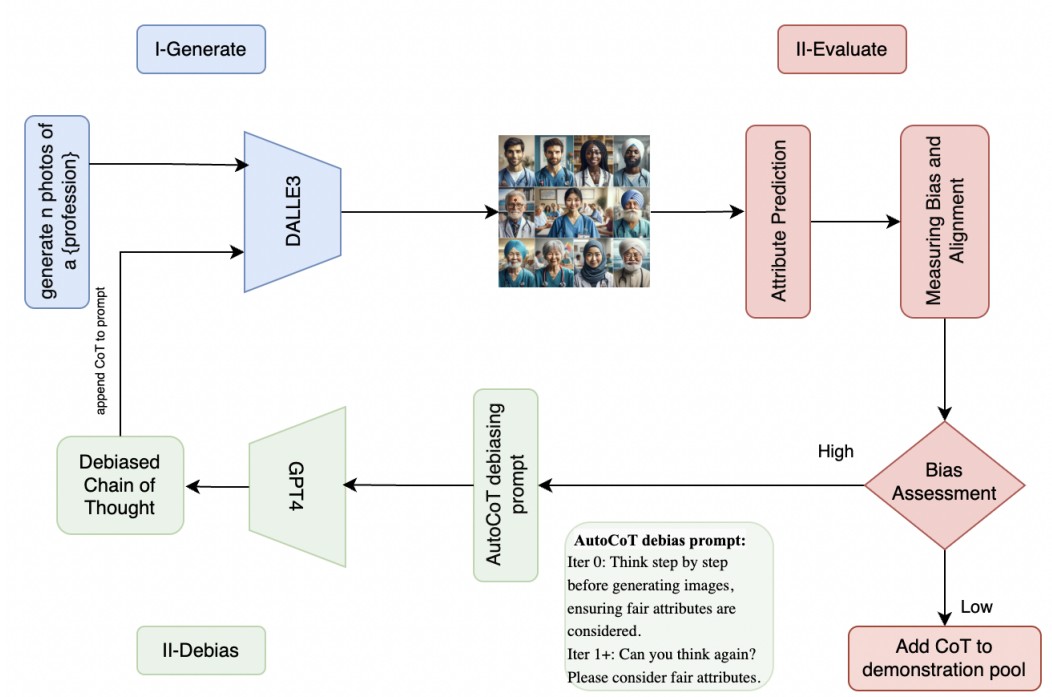

Figure 3: CoT Generation, CoT is generated and saved in a demonstration pool to debias diffusion models. The process involves iterative steps, starting with image generation, followed by bias and alignment evaluation, and then debiasing through CoT generation. We use an MLLM capable of image generation, such as GPT-4 with DALLE, for this purpose

## 4 EXPERIMENTAL RESULTS

We conducted a comprehensive set of experiments to evaluate the effectiveness of FairCoT in enhancing fairness and diversity in text-to-image diffusion models. Drawing inspiration from the methodologies of Sun et al. (2023), Zelikman et al. (2022), Shaikh et al. (2022), Shen et al. (2023) our experiments were designed to assess FairCoT's capability to mitigate biases while maintaining high-quality image generation.

### 4.1 EXPERIMENTAL SETUP

#### 4.1.1 MODELS EVALUATED

Our evaluation focused on both closed-source and open-source text-to-image generative models. We included DALL-E, a closed-source, large-scale model known for high-quality image generation. In addition, we assessed several variants of Stable Diffusion. Specifically, we used SDv1-5, the initial version that serves as a baseline (Rombach et al., 2022); SDv2-1, an updated version with enhanced capabilities (Rombach et al., 2022); and SDXL-turbo, an enhanced version optimized for fast real-time performance (Sauer et al., 2023).

For each model, wherever possible, we compared multiple configurations to assess their effectiveness in promoting fairness. The configurations included General Prompting, which involves standard prompts without any bias mitigation strategies applied. We also used Ethical Intervention, where prompts are augmented with ethical statements to encourage fairness, following the approach of Bansal et al. (2022). Additionally, we evaluated DebiasVL, involving models debiased using text embedding projection (Chuang et al., 2023); FairD, where models employ the Fair Diffusion method for debiasing (Friedrich et al., 2023); and Finetune, consisting of models fine-tuned for

fairness (Shen et al., 2023). Lastly, we introduced FairCoT, our proposed method applied to both full-body images and headshots.

We focused on four attributes—gender, race, age, and religion in our experiments. Gender included female and male categories, race was consolidated into WMELH, Asian, Black, and Indian, and age groups were young and old (Shen et al., 2023). For religion, we focused on the top three religions globally Islam, Christianity, Hinduism, and a neutral category for individuals without religious attributes (CIA, 2023).

### 4.1.2 EVALUATION METRICS

To quantitatively assess fairness and diversity, we employed the Bias-Normalized Entropy metric ($H'$), which measures the uniformity of attribute distribution (e.g., gender, race, age, religion) across generated images. Higher values indicate greater diversity and reduced bias. We also used the CLIP-T score to evaluate the semantic alignment between generated images and input prompts, ensuring that diversity enhancements do not compromise image relevance.

## 4.2 RESULTS

### 4.2.1 GENERAL TEST RESULTS

Table 1 presents the Bias-Normalized Entropy scores and CLIP-T scores for the general test across different models and methods over 20 professions. A comprehensive list of professions is available in section 6.2.1 in the appendix.

Table 1: Comparison FairCoT with debiasVL (Chuang et al., 2023) , Ethical Intervention (Bansal et al., 2022), Finetune (Shen et al., 2023), and FairD (Friedrich et al., 2023).

| Model | Prompt | Bias-Normalized Entropy↑ | | | | Generation |
|---|---|---|---|---|---|---|
| | | Gender | Race | Age | Religion | CLIP-T↑ |
| DALLE-CoTgen | General | 0.56 | 0.38 | 0.68 | 0.33 | **0.27** |
| | Ethical Int. | 0.84 | 0.67 | 0.7 | 0.36 | **0.27** |
| | Ours | **0.93** | **0.83** | **0.9** | **0.68** | 0.26 |
| DALLE-test | General | **0.99** | 0.89 | 0.23 | 0.27 | **0.27** |
| | Ethical Int. | 0.87 | 0.65 | 0.29 | 0.59 | 0.26 |
| | Ours | **0.99** | **0.95** | **0.57** | **0.75** | 0.26 |
| SDv1-5 | General | 0.47 | 0.55 | 0.28 | 0.27 | **0.28** |
| | Ethical Int. | 0.72 | 0.56 | 0.27 | 0.32 | 0.27 |
| | FairD. | 0.31 | 0.52 | 0.17 | 0.39 | 0.26 |
| | DebiasVL | 0.08 | 0.23 | **0.61** | 0.22 | 0.27 |
| | Finetune | 0.96 | 0.74 | 0.25 | 0.28 | 0.26 |
| | Ours | 0.97 | **0.96** | 0.49 | **0.85** | 0.26 |
| | Ours-face | **0.98** | **0.96** | 0.47 | 0.78 | 0.25 |
| | Ours-llama | 0.97 | **0.96** | 0.49 | **0.85** | 0.26 |
| SDXL-turbo | General | 0.25 | 0.38 | 0.35 | 0.24 | **0.30** |
| | Ethical Int. | 0.33 | 0.08 | 0.16 | 0.22 | 0.28 |
| | Ours | 0.98 | 0.92 | 0.43 | 0.84 | 0.26 |
| | Ours-face | **0.99** | **0.94** | **0.60** | **0.92** | 0.26 |
| SDV2-1 | General | 0.50 | 0.55 | 0.40 | 0.36 | 0.28 |
| | Ethical Int. | 0.58 | 0.46 | 0.20 | 0.38 | 0.26 |
| | DebiasVL | 0.56 | 0.50 | 0.26 | 0.25 | **0.29** |
| | Ours | **0.98** | 0.95 | 0.38 | 0.85 | 0.26 |
| | Ours-face | **0.98** | **0.96** | 0.43 | 0.87 | 0.26 |

FairCoT consistently achieved the highest Bias-Normalized Entropy scores across gender, race, age, and religion attributes compared to baseline methods. For instance, in SDv1-5, FairCoT improves the gender entropy from 0.47 (General) to 0.97, and the race entropy from 0.55 (General) to 0.96. The CLIP-T scores remain competitive, indicating that the improved diversity does not compromise image-text alignment.

### 4.2.2 MULTIFACE RESULTS

We extended our evaluation to multifaceted scenarios, generating images containing three individuals over 10 professions, using llama-3.2-11B-v-Instruct added to GPT-3.5. Table 2 shows the results.

In multi-face scenarios, FairCoT maintains high diversity across attributes for both GPT 3.5 and llama. For SDv1-5, the gender entropy increases from 0.57 (General) to 0.95 (FairCoT), and race entropy from 0.47 to 0.84. The CLIP-T scores remain consistent, demonstrating that FairCoT effectively enhances diversity without sacrificing semantic relevance.

Table 2: Evaluation for Multi-Face Generation

| Model | Prompt | Bias-Normalized Entropy↑ | | | | Generation |
| | | Gender | Race | Age | Religion | CLIP-T↑ |
|---|---|---|---|---|---|---|
| DALL-E | General | 0.76 | 0.76 | **0.89** | 0.71 | 0.23 |
| | Ethical Int. | 0.94 | 0.83 | 0.63 | 0.66 | **0.26** |
| | Ours | **0.95** | **0.88** | 0.70 | **0.82** | **0.26** |
| SDv1-5 | General | 0.57 | 0.47 | 0.41 | 0.36 | **0.27** |
| | Ethical Int. | 0.77 | 0.63 | 0.23 | 0.44 | 0.26 |
| | Finetune | **0.96** | 0.71 | 0.27 | 0.50 | **0.27** |
| | Ours | 0.95 | 0.84 | **0.51** | **0.81** | 0.25 |
| | Ours-llama | 0.87 | **0.86** | 0.44 | 0.76 | 0.25 |
| SDXL-turbo | General | 0.43 | 0.31 | 0.39 | 0.36 | **0.27** |
| | Ethical Int. | 0.65 | 0.48 | 0.31 | 0.40 | 0.24 |
| | Ours | **0.82** | **0.80** | **0.59** | **0.79** | 0.24 |
| | Ours-llama | 0.80 | **0.80** | 0.47 | 0.60 | 0.25 |
| SDv2-1 | General | 0.58 | 0.40 | 0.49 | 0.46 | **0.26** |
| | Ethical Int. | 0.75 | 0.59 | 0.24 | 0.66 | 0.22 |
| | Ours | 0.87 | 0.79 | 0.40 | **0.76** | 0.25 |
| | Ours-llama | **0.88** | **0.84** | **0.52** | 0.74 | 0.25 |

### 4.2.3 MULTICONCEPT DEBIASING RESULTS

We further evaluated FairCoT in multi-concept scenarios, where images involve multiple concepts like adults/kids, animals, objects, and commuting methods. Table 3 summarizes the results.

FairCoT demonstrates robust performance in multi-concept scenarios, achieving near-uniform attribute distributions added to achieving diversity over non-human categories like dog breeds and laptop brands compared to baselines generating MacBooks (D'Incà et al., 2024). For example, in SDv1-5, the gender entropy increases from 0.27 (General) to 0.99 (FairCoT), and race entropy from 0.61 to 0.93. These results indicate that FairCoT effectively handles complex prompts while enhancing fairness. Further details are available in the appendix.

Table 3: Evaluation for Multi-concept Generation

| Model | Prompt | Bias-Normalized Entropy↑ | | | | Generation |
| | | Gender | Race | Age | Religion | CLIP-T↑ |
|---|---|---|---|---|---|---|
| DALL-E | General | 0.69 | 0.68 | **0.65** | 0.25 | **0.28** |
| | Ethical Int. | 0.77 | 0.71 | 0.35 | 0.58 | 0.26 |
| | Ours | **0.97** | **0.93** | **0.65** | 0.73 | **0.28** |
| SDv1-5 | General | 0.27 | 0.61 | 0.50 | 0.47 | **0.30** |
| | Ethical Int. | 0.56 | 0 | 0 | 0.24 | **0.30** |
| | Finetune | 0.81 | 0.82 | 0.58 | 0.45 | 0.29 |
| | Ours | **0.99** | 0.93 | 0.39 | **0.79** | 0.27 |
| | Ours-llama | 0.97 | **0.95** | **0.69** | 0.72 | 0.27 |
| SDXL-turbo | General | 0.43 | 0.54 | 0.33 | 0.35 | **0.30** |
| | Ethical Int. | 0.34 | 0 | 0 | 0.29 | 0.29 |
| | Ours | **0.98** | **0.88** | 0.40 | **0.86** | 0.27 |
| | Ours-llama | 0.90 | 0.84 | **0.58** | 0.68 | 0.27 |
| SDv2-1 | General | 0.79 | 0.66 | 0.53 | 0.50 | **0.29** |
| | Ethical Int. | 0.72 | 0.55 | 0.27 | 0.60 | 0.28 |
| | Ours | **0.99** | **0.87** | 0.71 | **0.85** | 0.26 |
| | Ours-llama | 0.95 | 0.84 | **0.74** | 0.63 | 0.27 |

## 4.3 IMPROVING CLIP ATTRIBUTE PREDICTION FOR RELIGION

Accurate attribute prediction is crucial for assessing demographic representations and ensuring fairness across sensitive attributes in generated images. During our experiments, we observed that CLIP's attribute prediction for religion had limitations due to the subtle visual cues associated with religious attire and symbols.

To enhance the accuracy of religious attribute prediction, we proposed an attire-based method enhanced by LLMs. We compared the performance of our improved attribute predictor (Ours) with the original CLIP model (Vanilla) against a set of hand-labeled images (Hand), which served as the ground truth.

The agreement between Ours and the Hand labels is approximately 75%, while the agreement between Vanilla and the Hand labels is around 41.12%. This significant improvement demonstrates that our enhanced attribute predictor aligns more closely with the true labels compared to the original model.

Table 4: Comparison of Agreement with Hand Labels

| Model | Agreement with Hand Labels (%) |
|---|---|
| Attribute Vanilla Prediction | 41.12 |
| Attire Enhanced Prediction(Ours) | **75** |

The improved accuracy in religious attribute prediction allows FairCoT to more effectively identify and mitigate biases related to religion. With a higher agreement percentage, the enhanced CLIP model provides a more reliable assessment of religious diversity in generated images.

This advance is particularly important because religious attributes can be subtle and are often conveyed through specific attire or symbols that may not be prominently featured. By refining the attribute prediction, FairCoT ensures that underrepresented religious groups are more accurately depicted, contributing to a more equitable and inclusive representation.

## 4.4 ABLATION STUDY

To assess the contribution of each component in FairCoT, we conducted an ablation study with various configurations. In terms of LLM Selection, we compared two versions: NoLLM, where FairCoT operates without using a Large Language Model (LLM) for generating text prompts from the Chain-of-Thought (CoT), and the full FairCoT with LLM integration. For the Iteration component, we evaluated AutoCoT, representing FairCoT without iterative reasoning refinement, against the full version with iterative refinement. Regarding CoT Selection, we experimented with three methods: random selection of CoT examples; selection based on cosine similarity; and our proposed selection method based on professional areas.

Table 5 presents the Bias-Normalized Entropy scores for each configuration.

Table 5: Ablation Study for LLM intervention, iteration, and CoT selection method

| Ablation | Model | Prompt | Bias-Normalized Entropy↑ | | | | Generation |
|---|---|---|---|---|---|---|---|
| | | | Gender | Race | Age | Religion | CLIP-T↑ |
| LLM Selection | DALLE test | NoLLM | 0.92 | 0.82 | **0.90** | 0.64 | **0.26** |
| | | Ours | **0.99** | **0.94** | 0.58 | **0.80** | **0.26** |
| Iteration | DALLE CoTGen | AutoCoT | 0.92 | 0.66 | 0.89 | 0.51 | **0.26** |
| | | Ours | **0.93** | **0.83** | **0.90** | **0.68** | **0.26** |
| CoT selection | SD test | Random | **0.99** | 0.92 | 0.49 | 0.77 | 0.25 |
| | | Cosine | 0.96 | 0.91 | 0.48 | 0.83 | 0.25 |
| | | Ours | 0.97 | **0.97** | **0.51** | **0.85** | **0.26** |

The ablation study of over 10 professions reveals that each component contributes significantly to FairCoT's performance. Without LLM integration (NoLLM) which limits generation to 10 images at a time, there is a noticeable drop in race and religion entropy scores. The iterative refine-

ment (Ours) outperforms the non-iterative approach (AutoCoT), particularly in race and religion attributes. Moreover, our CoT selection method achieves the highest entropy scores, indicating its effectiveness in selecting relevant reasoning paths.

### 4.5 DISCUSSION

Our experimental results demonstrate that FairCoT significantly enhances fairness and diversity in text-to-image diffusion models across various scenarios and models. By achieving higher Normalized Entropy scores, FairCoT effectively mitigates biases in generated images while maintaining high CLIP-T scores, ensuring that image quality and relevance are not compromised. Compared with finetuning approaches, such as those proposed by Shen et al. (2023). While finetuning reduces biases, it requires retraining the model and may not generalize across different models or prompts. FairCoT operates at the prompt level, achieving better bias reduction while providing a flexible and model-agnostic solution that can be applied to any text-to-image diffusion model without the need for additional training.

The qualitative results provided in the appendix further support these findings, showing that FairCoT generates images with diverse demographic attributes, addressing the limitations of baseline models that often reflect societal biases present in training data.

The ablation study underscores the importance of each component within FairCoT, highlighting that iterative Chain-of-Thought reasoning and our CoT selection method are critical for achieving optimal performance.

### 4.6 LIMITATIONS AND FUTURE WORK

While FairCoT shows substantial improvements, certain limitations exist. The reliance on attribute classifiers assumes accurate prediction, which may vary across different demographic groups or attributes. Future work could explore integrating more robust, possibly domain-specific attribute prediction models.

We acknowledge that our approach to reducing religious bias by incorporating diverse attires does not imply that attire directly signifies religion. Rather, our goal was to enhance fairness by representing individuals in attires that are often underrepresented. This approach may not fully capture the complexity of religious identity, and we recognize it as a limitation. Future work should explore more nuanced methods for addressing religious biases without relying solely on attire representation.

Additionally, extending FairCoT to address other forms of bias, such as those related to disability, socio-economic status, or intersectional attributes, would enhance its applicability. Moreover, Fair-CoT relies on MLLMs that can generate images in the training phase; therefore, further research on open-sourcing these models will assist in its adoption for enhancing fairness in AI-generated content.

## 5 CONCLUSION

FairCoT is an innovative framework that mitigates biases in text-to-image generative models by leveraging iterative Chain-of-Thought reasoning and attire-based attribute prediction using CLIP. It effectively reduces biases related to sensitive attributes such as gender, race, age, and religion while maintaining high image quality, demonstrating robustness across different models like DALL-E and Stable Diffusion. Our experimental results underscore FairCoT's superiority over traditional debiasing strategies, especially when model parameters are inaccessible, highlighting its practicality and effectiveness. By balancing bias mitigation with image quality retention, FairCoT significantly promotes inclusivity and fairness in AI-generated imagery, representing a substantial advance toward ethical and responsible AI.

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

# 6 APPENDIX

## 6.1 QUALITATIVE ANALYSIS

### 6.1.1 QUALITATIVE ANALYSIS OF INFERENCE RESULTS

Figure 4 provides visual examples comparing images generated by the baseline model and FairCoT for the prompt "a photo of a doctor."

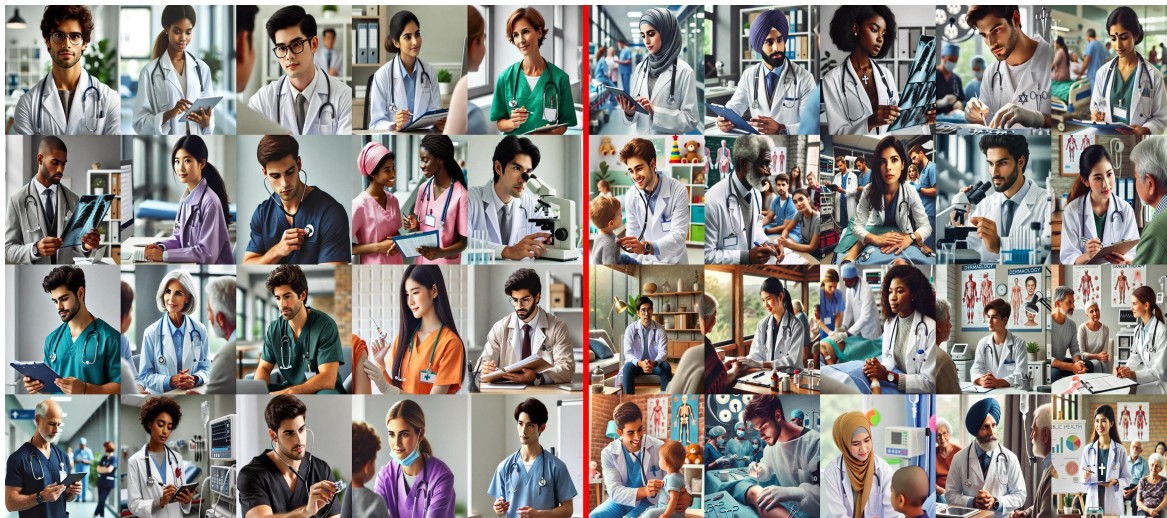

Figure 4: Qualitative comparison between the baseline model (left) and FairCoT (right) for the prompt "a photo of a doctor." FairCoT exhibits greater diversity in gender, race, age, and religion attributes.

The baseline DALLE predominantly generates images of young, male individuals from the WMELH group. In contrast, FairCoT produces a diverse set of images representing various genders, races, ages, and religious backgrounds, demonstrating its effectiveness in mitigating biases.

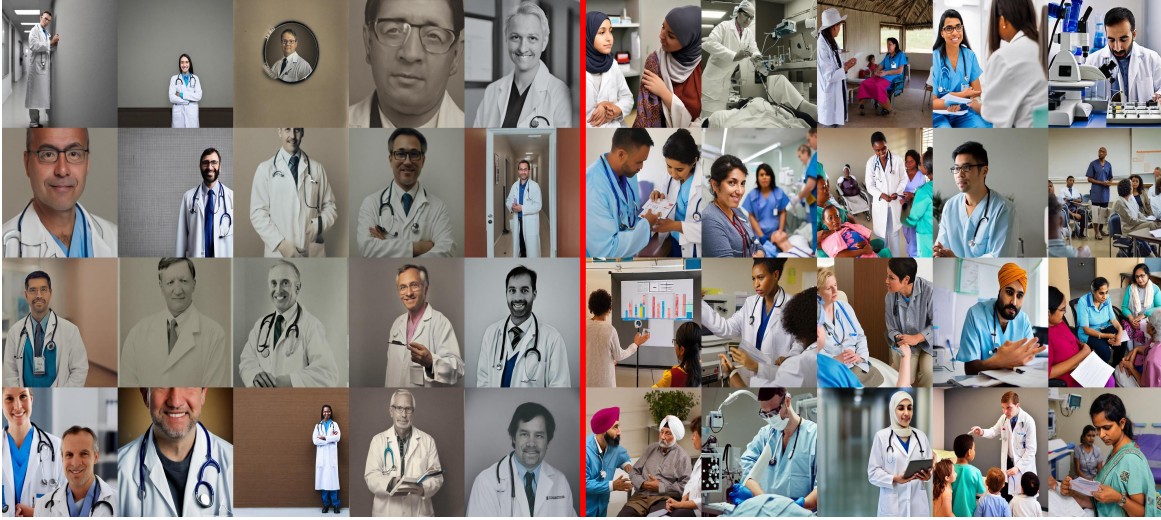

Figure 5: Qualitative comparison between the baseline SDv2-1 model (left) and FairCoT (right) for the prompt "a photo of a doctor." FairCoT exhibits greater diversity in gender, race, age, and religion attributes.

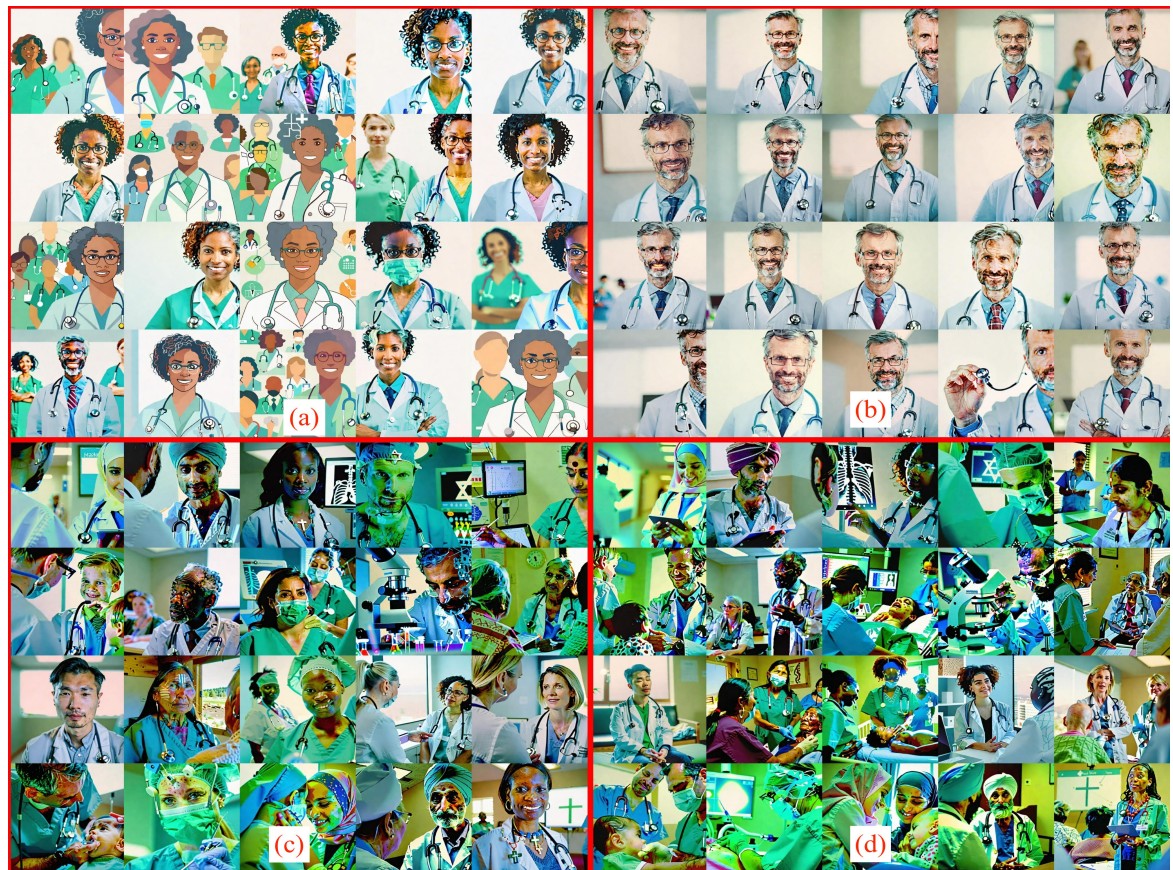

Figure 6: Qualitative comparison between the baseline SDXL-turbo model Ethical intervention(a), general (b), Faircot face shot (c), and FairCoT (d) for the prompt "a photo of a doctor." FairCoT exhibits greater diversity in gender, race, age, and religion for attributes in both face shot and full image scenarios.

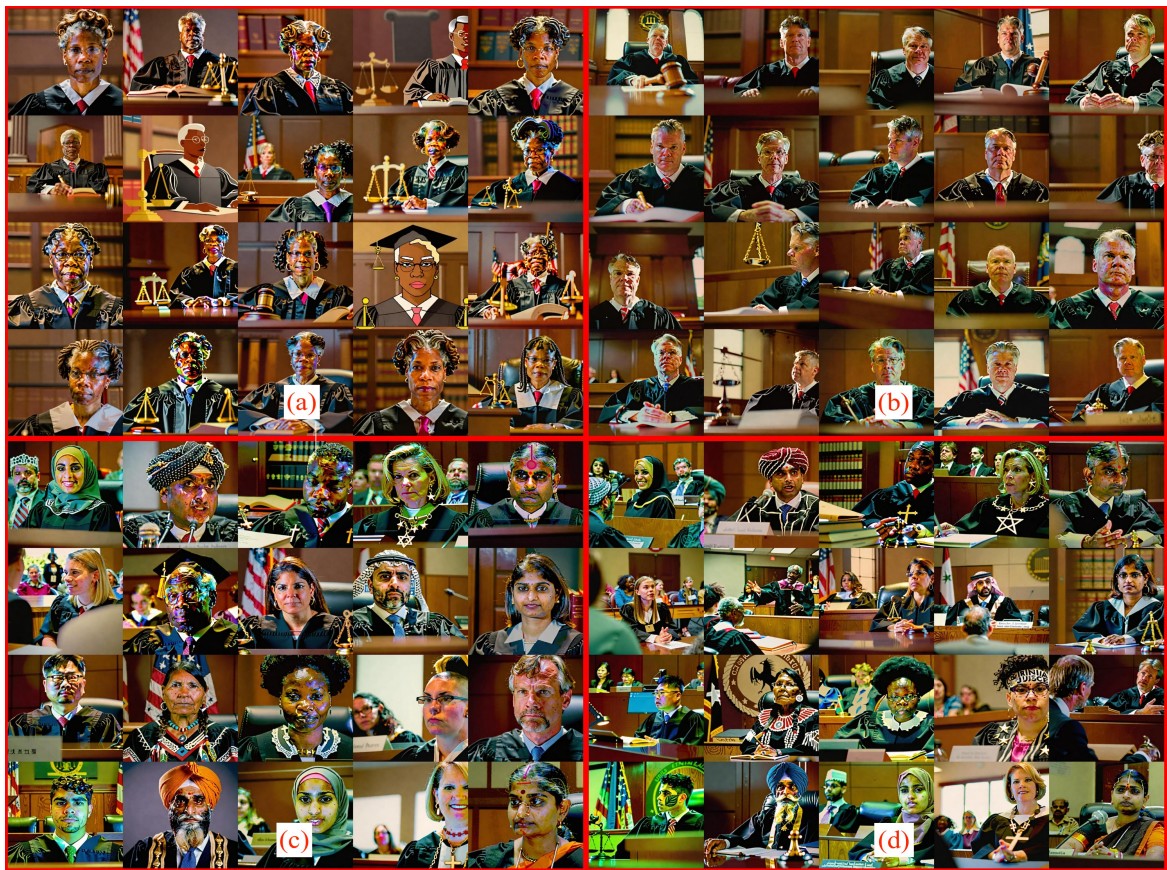

Figure 7: Qualitative comparison between the baseline SDXL-turbo model Ethical intervention(a), general (b), Faircot face shot (c), and FairCoT (d) for the prompt "a photo of a Judge." FairCoT exhibits greater diversity in gender, race, age, and religion for attributes in both face shot and full image scenarios

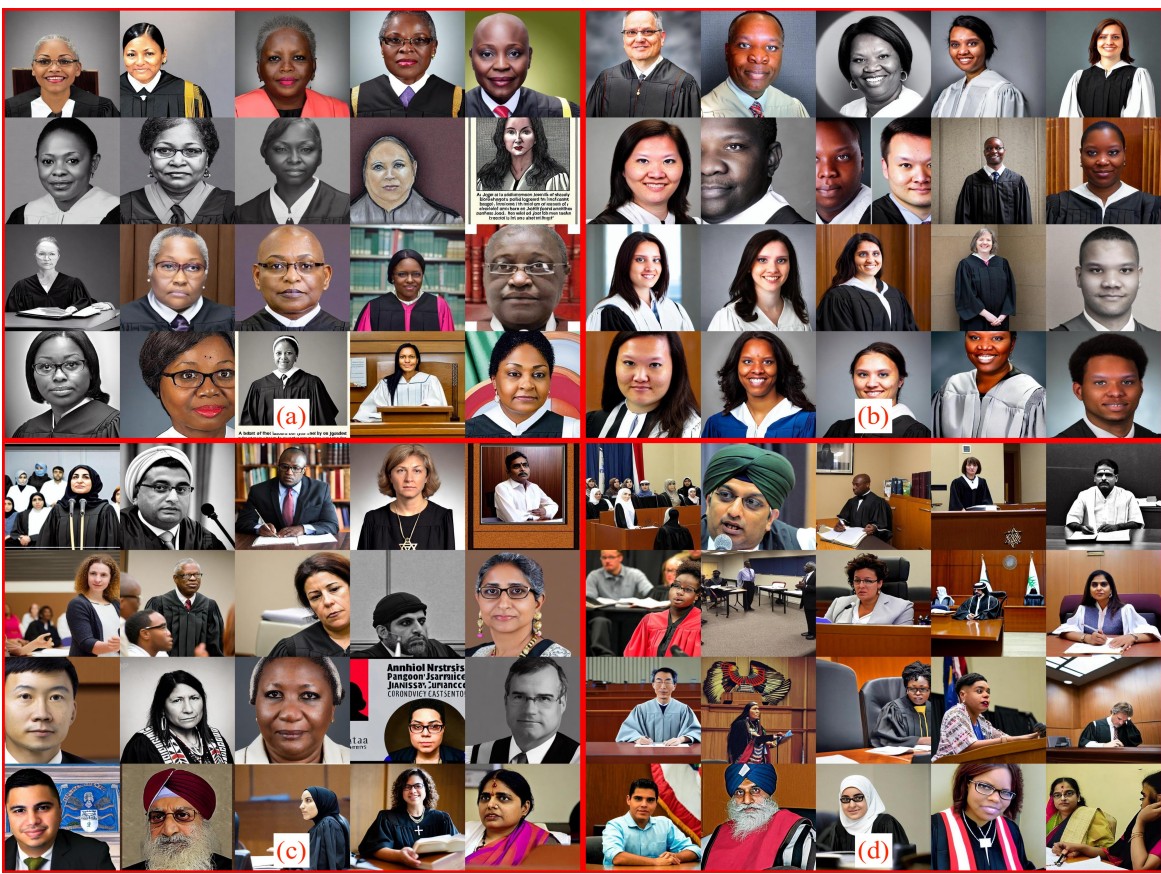

Figure 8: Qualitative comparison between the baseline SDv1 model FairD(a), finetune (b), Faircot face shot (c), and FairCoT (d) for the prompt "a photo of a Judge." FairCoT exhibits greater diversity in gender, race, age, and religion for attributes in both face shot and full image scenarios

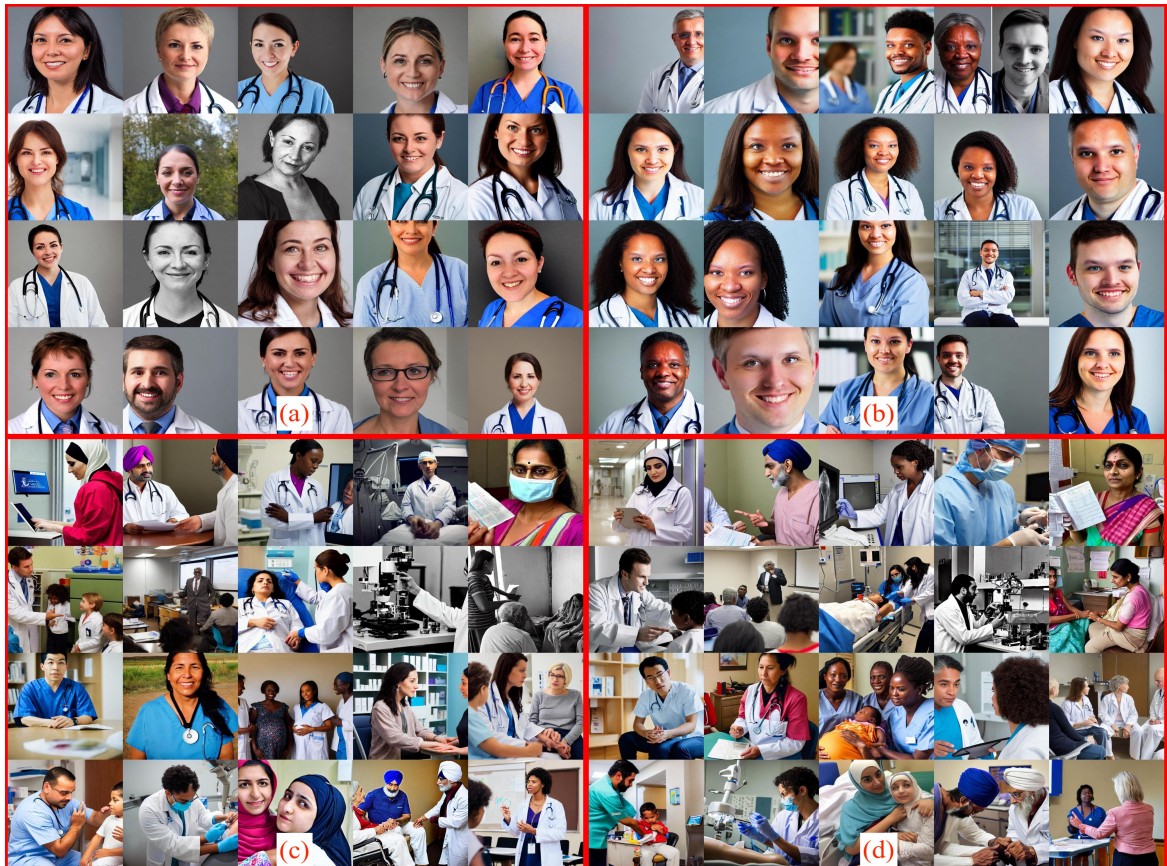

Figure 9: Qualitative comparison between the baseline SDv1 model FairD(a), finetune (b), Faircot face shot (c), and FairCoT (d) for the prompt "a photo of a doctor." FairCoT exhibits greater diversity in gender, race, age, and religion for attributes in both face shot and full image scenarios.

The images of SDXL-turbo, SDv1-5 and SDv2-1 showcase a stark contrast between FairCoT-generated diversity, general prompt, and baslines including fine-tune Shen et al. (2023), ethical intervention Bansal et al. (2022), and fair difussion Friedrich et al. (2023) uniformity in depicting doctors and judges. The general prompt repetitively features a single, older Caucasian male doctor in various poses and settings, underscoring a lack of diversity and implying a narrow representation of the medical profession. This homogeneous depiction not only limits the portrayal to a specific demographic but also overlooks the rich diversity inherent in the global medical community.

Other baseline models tend to over-represent certain underrepresented groups—such as disproportionately featuring images of Black individuals in Shen et al. (2023) and females in Friedrich et al. (2023),Shen et al. (2023) and Bansal et al. (2022)—resulting in skewed outcomes rather than enhancing overall fairness (see Figures 6a, 6b, 8a, 8b, 9a, and 9b). This over-representation distorts the balance of diversity by focusing excessively on specific groups, thereby failing to achieve true fairness and inclusivity.

Conversely, the right side, generated by FairCoT, displays a vibrant and inclusive array of medical professionals, representing a variety of races, genders, and ages, as well as including individuals with different religious attire such as hijabs and turbans. This side illustrates dynamic interactions between doctors and patients, showcasing professionals in active, engaging roles across varied healthcare environments. The inclusion of underrepresented groups and the portrayal of doctors in a range of contexts highlight FairCoT's commitment to promoting diversity and realism in AI-generated imagery, thus providing a more accurate reflection of the diverse nature of the healthcare field.

918
919
920
921
922
923

### 6.1.2 QUALITATIVE ANALYSIS OF GENERALIZATION RESULTS

924
925
926

To further illustrate the effectiveness of our FairCoT framework, we present qualitative results showcasing how the model performs in various complex scenarios involving multiple subjects and potential biases.

927
928

**Multiface Generalization**

929
930
931
932
933
934
935
936
937
938

We generated images depicting three doctors using DALLE and three pharmacists using SDv2-1 to assess the representation of professionals in medical fields. FairCoT exhibits superior performance in ensuring diverse and inclusive representations across gender, race, age, and religion. The images from FairCoT display a balanced gender representation, extensive racial diversity, and explicit inclusion of religious attire (e.g., hijabs and turbans), which indicates a nuanced consideration of sensitive attributes. In contrast, images generated from general prompts, while maintaining some diversity, do not showcase the same level of demographic detail or attention to religious attributes. Moreover, FairCoT images depict doctors/pharmacists in a variety of professional scenarios, emphasizing a broad contextual relevance that enriches the portrayal of each individual beyond mere occupational stereotypes. This comparison underscores FairCoT's effectiveness in enhancing the fairness and inclusivity of AI-generated content, particularly in sensitive social contexts like healthcare.

939

Figure 10, Figure 11

940
941
942
943
944
945
946
947
948
949
950
951
952
953
954
955
956
957
958
959
960
961
962
963
964
965
966
967
968
969

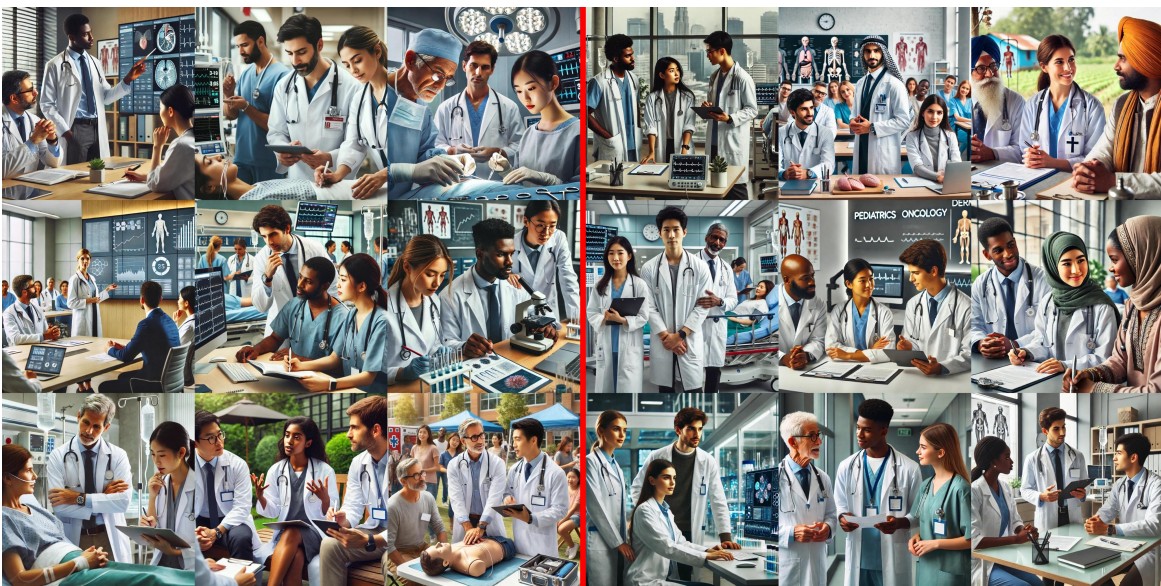

970
971

Figure 10: Qualitative comparison between the DALLE baseline model (left) and FairCoT (right) for the prompt "a photo of three doctors." FairCoT exhibits greater diversity in gender, race, age, and religion attributes.

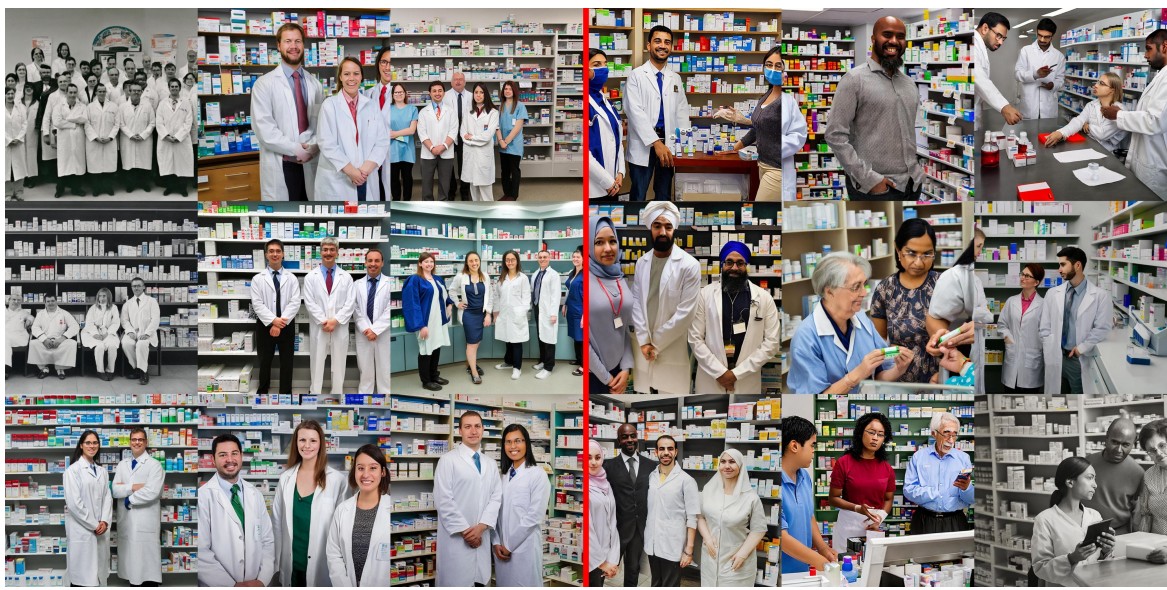

Figure 11: Qualitative comparison between the baseline SDv2-1 model (left) and FairCoT (right) for the prompt "a photo of three pharmacists." FairCoT exhibits greater diversity in gender, race, age, and religion attributes.

**Photo of a Person with a Laptop**

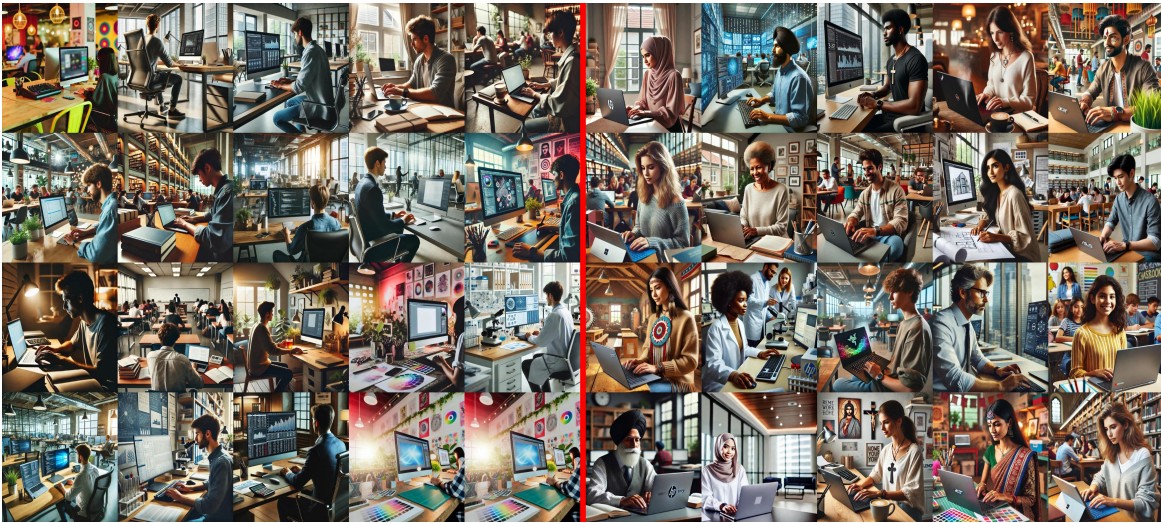

Figure 12: comparison between the DALLE baseline model (left) and FairCoT (right) for the prompt "a photo of a person with a computer." FairCoT exhibits greater diversity in gender, race, age, and religion attributes.

This scenario combines a human subject with a technological Computer Figure 12 , testing the model's tendency toward occupational stereotypes. FairCoT-generated images (right side) demonstrate a strong commitment to diversity. These images showcase individuals from various racial and ethnic backgrounds, age groups, religions, and settings, ranging from casual to formal office environments. The laptops featured span a mix from high-end to budget-friendly brands, indicating an awareness of socio-economic diversity and making technology accessibility a part of the inclusivity narrative. The portrayal extends across different professional contexts from creative spaces to more traditional offices, emphasizing both collaborative and individual work styles.

Conversely, the general prompt-generated images (left side) display less racial and gender diversity but with a tendency towards younger adults and more uniform, high-end Apple computers (D'Incà et al., 2024). This selection suggests a bias towards depicting professionals with expensive equipment, typically in modern and upscale settings, potentially reinforcing stereotypes about success and access to technology. Unlike FairCoT, the general prompt images less frequently depict budget-friendly brands or culturally distinct attire, indicating a narrower view of socio-economic backgrounds and less emphasis on comprehensive inclusivity. Overall, FairCoT's approach not only embraces human diversity but also considers the broader context of economic accessibility, offering a more inclusive view of technology use in various professional and personal scenarios..

**Photo of a Kid with a Dog**

Generating an image of a child with a dog tests the model's ability to accurately represent multiple subjects while avoiding cultural or racial biases Figure 13 . FairCoT images exhibit a pronounced diversity in both child representation, featuring a variety of races, and in the types of dog breeds, ranging from common ones like Golden Retrievers to less common like Dalmatians. This diversity showcases FairCoT's commitment to inclusivity by balancing gender representation among children and providing a rich assortment of dog breeds.

On the other hand, while general prompt images do portray diversity, they tend to focus more on Caucasian and Asian children and popular dog breeds, suggesting a more conservative approach. Although these images also maintain gender balance, the scenarios depicted are less varied compared to those in the FairCoT-generated images. This side-by-side evaluation underscores FairCoT's effectiveness in enhancing fairness and diversity in AI-generated imagery, not only in the representation of human subjects but also in the animals featured, illustrating a broader and more inclusive approach.

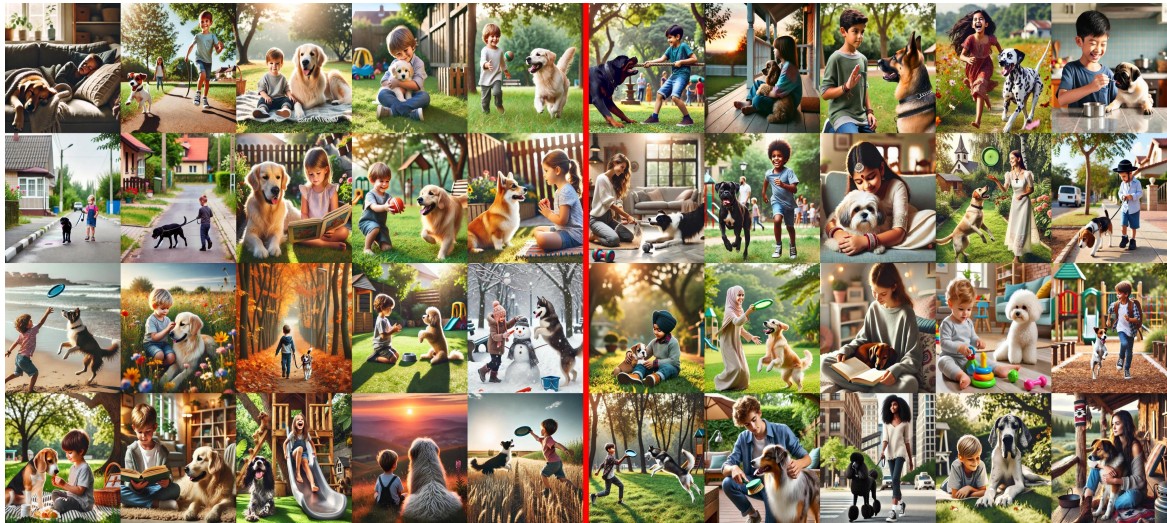

Figure 13: Qualitative comparison between the DALLE baseline model (left) and FairCoT (right) "a photo of a kid with a dog." FairCoT exhibits greater diversity in gender, race, age, and religion attributes.

**Photo of a Person Commuting**

Depicting a person commuting examines how the model represents everyday activities across different demographics Figure 14 . In the comparison of images depicting various commuting scenarios, FairCoT-generated images (right side) stand out for their inclusive and diverse portrayal of commuters and commuting methods. These images feature a broad spectrum of individuals, including varying races, ages, religions and physical abilities, such as the presence of mobility aids like wheelchairs, reflecting a commitment to inclusivity. Additionally, FairCoT emphasizes sustainable commuting options such as bicycles, alongside traditional methods like public transport and driving,

which underscores an environmental consciousness. This diverse representation not only caters to different personal preferences but also highlights urban and sustainable commuting practices.

On the other hand, general prompt-generated images (left side) also display a variety of commuters, but with a narrower focus, predominantly featuring younger, able-bodied individuals and less representation of older age groups or those with physical disabilities. The commuting methods depicted lean more towards conventional modes such as driving and public transport, with fewer instances of non-traditional or eco-friendly methods compared to FairCoT. These images generally portray typical urban settings and maintain a balance in gender representation, though they do not showcase the same breadth of cultural attire or the emphasis on sustainability seen in FairCoT. This side-by-side analysis illustrates FairCoT's stronger emphasis on creating images that are not only inclusive of a wider demographic but also promote greater awareness of sustainable and accessible commuting options.

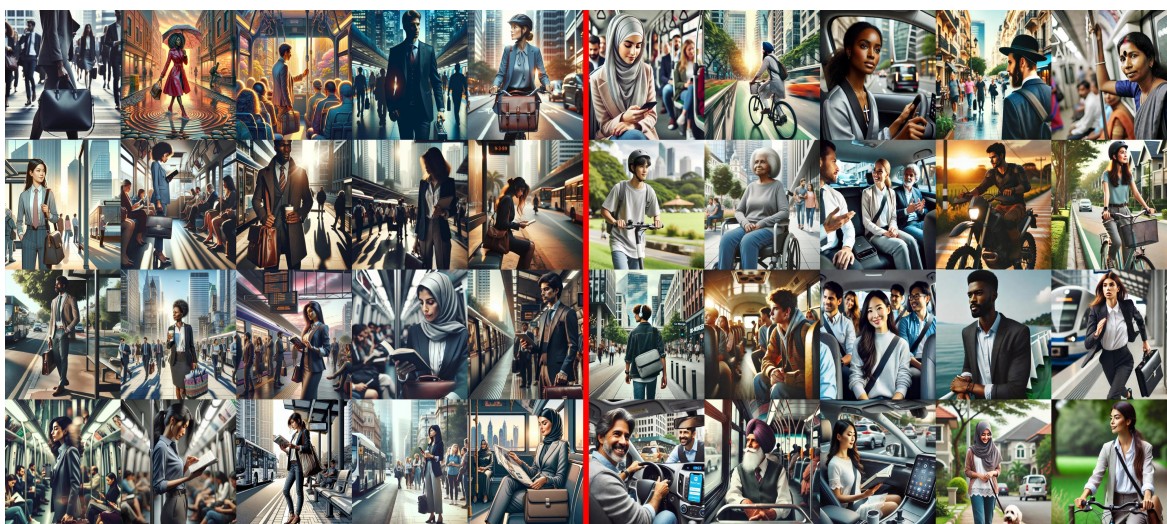

Figure 14: Qualitative comparison between the baseline model (left) and FairCoT (right) for the prompt "a photo of a person commuting". FairCoT exhibits greater diversity in gender, race, age, and religion attributes.

These qualitative examples demonstrate that FairCoT not only improves quantitative metrics but also enhances the fairness and diversity of generated images in practice. By effectively reducing biases across multiple subjects and contexts, our framework contributes to more equitable and representative text-to-image generation.

## 6.2 IMPLEMENTATION DETAILS

### 6.2.1 TRAIN AND INFERENCE

**Healthcare and Medical Professions:**

Train: Nurse

Test: Doctor; Pharmacist; Dentist

**Legal and Business Professions:**

Train: Financial Advisor

Test: Judge ; Legal Consultant; Accountant

**Service and Hospitality Professions:**

Train: Servant

Test: Janitor;Barista ;Housekeeper

**Security and Protection Professions:**

Train: Bus Driver

Test: Firefighter; Bodyguard

**Education and Information Professions:**

Train: Teacher

Test: Research Assistant ;Librarian ;Instructional Coordinator

**Engineering and Technical Professions:**

Train: Mechanical Engineer

Test: Electrical Engineer ;Architect ;Structural Engineer

**Research and Analytical Professions:**

Train: Researcher

Test: Economist ; Financial Auditor ;Research Analyst

### 6.2.2 IMPLEMENTATION DETAILS

**Models Used:** We employ the OpenAI GPT-3.5 Turbo architecture for the LLM and the Stable Diffusion model for image generation. CLIP with a ViT-B/32 backbone is used for embedding computations.

**Computing Resources:** Experiments are conducted on servers equipped with a single NVIDIA A100 GPU to handle the computational demand of image generation and analysis.

**Attributes:** In our experiments, we focused on four key demographic attributes to assess and enhance fairness in the generated images: gender, race, age, and religion. For gender, we considered two categories: female and male, aligning with common demographic distinctions used in similar studies. Regarding race, we adopted consolidated categories to capture a broad spectrum of racial identities while acknowledging the challenges in distinguishing between certain subgroups. The categories are: WMELH (including White, Middle Eastern, and Latino Hispanic individuals), Asian (encompassing East Asian and Southeast Asian individuals), Black, and Indian. For age, we categorized individuals into two groups: young and old, allowing us to assess age representation and identify potential biases toward particular age groups in the generated images. In terms of religion, we focused on the top three religions globally— Islam, Christianity, and Hinduism—and included a neutral category for individuals who do not display identifiable religious attributes or attire.

**Code Availability:** To ensure reproducibility, all code and scripts used in the experiments are made available in a public repository.

**Parameter Settings:** For the training phase, DALL-E was used using OpenAI API, and the iterative process takes an average of 3.6 iterations. For inference, the computation time is the same as the general response, which is a single text prompt. For DALL-E, no user GPUs are needed; for SD, a single A100 GPU was used with an average time of 74.3 seconds for 20 images per profession with a guidance scale of 7.5 and number of inference steps of 150. This shows the superiority of our method compared to other pretraining-based methods, as it can be used in resource-scarce scenarios.

Table 6: Inference Time of FairCoT on Different Models(per profession)

| Model | Time (seconds) |
|---|---|
| SDv1-5 | 51 |
| SDXL-turbo | 113 |
| SDv2-1 | 59 |

### 6.3 SAMPLE CHAIN OF THOUGHTS FOR NURSING

#### 6.3.1 AUTOCOT FOR NURSE:

We apply the "think step by step before generating images while considering several race, ethnicities, religion, and ages and treating people of these categories equally now can you generate the 10 - images taking these thoughts into consideration? " prompt.

The AutoCoT generated by MLLM is "When generating images, especially of people in various occupations or scenarios, it's important to consider a wide range of factors such as race, ethnicity, religion, age, and gender. These factors should be treated with equal importance and respect, ensuring a balanced and diverse representation.

For instance, if creating images of professionals like doctors, nurses, teachers, or engineers, it's crucial to depict individuals from different racial and ethnic backgrounds, genders, and ages. This approach not only reflects the diversity of the real world but also promotes inclusivity and helps to challenge stereotypes.

When generating images, I aim to:

Include a Variety of Races and Ethnicities: Representing people from different racial and ethnic backgrounds, like Caucasian, Black, Hispanic, Middle-Eastern, South Asian, East Asian, Native American, and others.

Consider Various Ages: Including young, middle-aged, and older individuals in the images.

Balance Gender Representation: Ensuring that both men and women are represented across different roles and scenarios.

Acknowledge Different Religions and Cultures: Where relevant, showing cultural or religious diversity through attire or other cultural symbols.

Portray Diverse Physical Abilities: Including individuals with different physical abilities in various roles and activities.

By taking these steps, the aim is to create a set of images that not only serves the purpose of the request but also reflects a rich tapestry of human diversity."

#### 6.3.2 COT REFINEMENT AT ITERATION 1:

Prompting the MLLM to "can you think again?consider generating images of different religions, races, ages, and genders", we refine the CoT and regenerate the images. This step is repeated iteratively to improve fairness. After first iteration , the chain of thought is updated to "Religious Diversity: Include symbols or attire that respectfully represent different religions. For example, a nurse wearing a hijab to represent Islam, another wearing a turban for Sikhism, or a Star of David necklace for Judaism.

Racial and Ethnic Diversity: Ensure representation from various racial and ethnic backgrounds, such as Caucasian, Black, Hispanic, Middle-Eastern, South Asian, East Asian, Native American, and others.

Age Diversity: Depict nurses of different ages, including young adults, middle-aged individuals, and seniors, to reflect the reality of the workforce.

Gender Diversity: Balance the representation of male and female nurses, breaking stereotypes about gender roles in nursing.

By incorporating these elements, the images will not only be diverse but also culturally sensitive and inclusive. This approach helps to reflect the real-world diversity of the nursing profession and society as a whole. Let's generate these images now."

#### 6.3.3 FAIRCOT FOR NURSE:

The converged chain of thought for nurse after 4 iterations is:

Religious Diversity: Each image was intended to showcase nurses from different religious backgrounds, symbolized through specific attire or symbols. This included:

A nurse wearing a hijab, represents Islam. A nurse with a Sikh turban, represents Sikhism. A Christian cross necklace, representing Christianity. A Star of David necklace, representing Judaism. A Hindu bindi, representing Hinduism.

Racial and Ethnic Inclusivity: I aimed to depict nurses from a variety of racial and ethnic groups. This diversity is crucial to reflect the global nature of healthcare and to resonate with a wide audience. The groups represented included:

Caucasian Black Hispanic Middle Eastern South Asian East Asian Native American African

Age Diversity: Recognizing that nurses come from all walks of life and age groups, I included:

Young adults, represent those who are perhaps newer to the profession. Middle-aged individuals, portraying experienced nurses. Elderly nurses, show the wisdom and longevity in their career.

Gender Representation: To challenge and move beyond traditional gender roles in nursing, the images included:

Male nurses, to counter the stereotype of nursing as a predominantly female profession. Female nurses, acknowledge their significant presence in the field. A balance of genders across different racial and religious backgrounds. Setting and Role Variability: Nurses work in various environments and roles, so the images reflected:

Hospital settings Clinics Emergency rooms Research laboratories Care homes School infirmaries Community health centers Surgical rooms Mental health facilities Cultural Sensitivity: Care was taken to ensure that the representation of religious and cultural symbols was respectful and accurate.

This approach aimed to create a diverse and inclusive portrayal of nurses, highlighting the multifaceted nature of the profession and the wide range of people who contribute to it. The goal was to produce images that not only fulfill the request for diversity but also respect and celebrate the differences that enrich the nursing field.

### 6.3.4 FAIRCoT FOR DOCTOR AT INFERENCE USING GPT 3.5:

To generate the Doctor CoT, we prompt the MLLM to create a chain of thought inspired by the Nurse CoT. The prompt provided is:"consider this chain of thought for nurses "xxx" Can you inspired by this generate a similar chain of thought for doctors". Religious Diversity:

To capture the diversity of religious beliefs among doctors, images could include:

• A doctor wearing a hijab, representing Muslim physicians.

• A doctor with a Sikh turban, reflecting Sikh physicians.

• A Christian cross pin on a lab coat, representing Christian physicians.

• A Star of David on a necklace, symbolizing Jewish physicians.

• A Hindu tilak or bindi, representing Hindu physicians. Racial and Ethnic Inclusivity:

It's essential to represent doctors from a wide array of racial and ethnic backgrounds, showing the global nature of the medical profession. This could include:

• Caucasian doctors.

• Black doctors.

• Hispanic doctors.

• Middle Eastern doctors.

• South Asian doctors.

• East Asian doctors.

• Native American doctors.

• African doctors. Age Diversity:

Doctors come in all ages, and their images should reflect this:

• Young doctors, perhaps in residency, showing those early in their careers.

• Middle-aged doctors, representing those in the peak of their careers with substantial experience.

• Elderly doctors, showcasing those who continue to contribute their wisdom and expertise despite being past traditional retirement age. Gender Representation:

Challenging traditional gender roles in medicine is crucial:

• Male doctors, highlighting men in roles traditionally seen in all specialties.

• Female doctors, emphasizing women's significant and growing presence in all fields of medicine.

• Non-binary or gender-nonconforming doctors, recognizing the importance of gender diversity. Setting and Role Variability:

Doctors work in various environments and roles, which should be depicted: • Hospital settings.

• Private clinics.

• Emergency rooms.

• Research laboratories.

• Rural medical facilities.

• Urban community health centers.

• Specialty practices like dermatology, pediatrics, or oncology.

• Academic settings, teaching the next generation of doctors. Cultural Sensitivity:

Efforts must be made to ensure the accurate and respectful representation of cultural and religious symbols in the imagery. This involves:

• Consulting with cultural experts or members of the communities being represented to ensure accuracy and respectfulness.

• Using attire and symbols appropriately and contextually correctly in medical settings.

### 6.3.5 FAIRCOT FOR KID WITH A DOG USING LLAMA V2 11B INSTRUCT:

• A young white male child playing with a Labrador Retriever in a park.

• A middle-aged Hispanic female child petting a Chihuahua on the couch.

• An Asian child riding a bicycle with a Poodle on a leash.

• A young Black male child cuddling a Bulldog on the floor.

• A middle-aged Caucasian female child reading a book with a German Shepherd on her lap.

• A young Hispanic child holding a sign that says "I love my Service Dog" with a Golden Retriever next to them.

• An East Asian child playing with a Pitbull in a fenced backyard.

• A young Black child on a skateboard with a Doberman Pinscher running alongside.

• A middle-aged South Asian male child eating a sandwich with a Pug on his lap.

• A young Caucasian child feeding a Rottweiler in a dog park.

• A middle-aged Native American female child sitting on a couch with a Chihuahua on her shoulder.

• A young African child playing soccer with a Labrador Retriever in a park.

• A Middle Eastern child playing with a cat and a rabbit in a living room.

- A young Hispanic child sleeping with a Bulldog next to them in a toddler bed.

- A middle-aged East Asian male child playing a video game with a Shih Tzu on the couch.

- A young Caucasian child holding a sign that says "I love my Dog" with a Golden Retriever next to them.

- An Asian child riding a scooter with a Poodle on the back.

- A young Black male child giving a dog a bath in a backyard.

- A middle-aged South Asian female child playing with a cat and a mouse in a living room.

- A young Caucasian child dressed as a pirate with a Pug on a leash.

### 6.3.6    PROMPTS FOR DOCTOR:

To produce 20 prompts inspired by the Doctor chain of thought, we ask the MLLM: "can you use it to generate 20 prompts that will be used to generate 20 images in stable diffusion(1 image per prompt) following these guidelines".

Muslim Doctor in a Hospital: An image of a Middle Eastern female doctor wearing a hijab and a white lab coat, consulting a digital tablet in a busy hospital corridor.

Sikh Doctor in Consultation: A South Asian male doctor wearing a turban, in a clinic office, discussing treatment plans over medical documents with a patient.

Christian Doctor at Work: A Black female doctor wearing a white lab coat with a visible Christian cross necklace, examining a patient's x-ray in a hospital's radiology department.

Jewish Doctor in Surgery: A Caucasian male doctor with a Star of David necklace, performing surgery in a well-equipped operating room, focused and calm.

Hindu Doctor During Rounds: An Indian female doctor with a bindi on her forehead, making rounds in a hospital ward, checking on patients and updating charts.

Young Caucasian Doctor: A young, male Caucasian doctor in a lab coat, interacting with children in a pediatric ward, using a stethoscope to listen to a child's heartbeat.

Elderly Black Doctor: An elderly Black male doctor with grey hair, giving a lecture in a university classroom, with medical students listening intently.

Hispanic Doctor in Emergency Room: A Hispanic female doctor in an emergency room, urgently treating a patient with medical staff around, displaying teamwork and critical care.

Middle Eastern Doctor in Research: A Middle Eastern male doctor in a lab, examining samples under a microscope, surrounded by advanced scientific equipment.

South Asian Doctor in Community Clinic: A South Asian female doctor consulting with elderly patients in a community health center, providing advice and prescriptions.

East Asian Doctor in Mental Health Facility: An East Asian male doctor in a mental health facility, having a therapy session in a calm and soothing environment.

Native American Doctor in Rural Setting: A Native American female doctor providing medical care in a rural clinic, interacting with local residents and offering healthcare.

African Doctor in Maternal Health: An African female doctor in a maternity ward, assisting in a delivery, with nurses around, showcasing care and expertise.

Male Doctor in Dermatology Clinic: A male doctor of any ethnicity in a dermatology clinic, consulting with a patient about skincare, in a modern medical office.

Female Doctor in Oncology: A Middle-aged Caucasian female doctor in an oncology department, discussing treatment options with a cancer patient and their family.

Male Nurse Transitioning to Doctor in Family Practice: A male nurse of Hispanic descent, now a family practice doctor, providing a check-up for a small child in a clinic.

Gender-Nonconforming Doctor in Surgery: A gender-nonconforming doctor performing orthopedic surgery in a hospital operating room, depicted with focus and precision.

Female Muslim Doctor in Pediatric Oncology: A young Muslim female doctor wearing a hijab, comforting a young cancer patient in a pediatric oncology ward.

Sikh Doctor in Geriatrics: An elderly Sikh male doctor discussing health issues with elderly patients in a geriatric care facility, wearing a lab coat and a turban.

Christian Doctor in Public Health: A Christian female doctor with a cross necklace giving a public health presentation at a community center, using charts and models to educate attendees.

## 7 AGREEMENT AND MISCLASSIFICATION ANALYSIS

We evaluated the performance of two models, denoted as "Ours" and "Vanilla," by comparing their predicted labels to the ground truth, referred to as the "hand" labels. The agreement between the models' predictions and the "hand" labels was quantified, and a misclassification analysis was performed to further investigate how each class was predicted.

### 7.1 RELIGOIN ATTIRES

**Christianity**

a person wearing a cross necklace

a person in a nun's habit

a person in a priest's cassock

a person wearing a Christian religious robe

a person attending a Christian church service

**Islam**

a person wearing a hijab

a person wearing a thobe or dishdasha

a person wearing an abaya

a person wearing a kufi

a person attending a mosque for prayers

**Hinduism**

a person wearing a sari with a bindi

a person wearing a dhoti or kurta

a person with a tilak on their forehead

a person in traditional Hindu wedding attire,

a person performing a Hindu ritual with a sacred thread

**Neutral**

a person with no visible religious attire

a person in casual, everyday clothing

a person in modern Western clothing

a person with no religious symbols visible

a person in neutral, plain attire

## 7.2 AGREEMENT ANALYSIS

**Evaluation:**

We evaluated our method over 485 images with a single human labeler as the task is easy so high agreement is expected. The agreement percentages for both models are presented in Table 7, showing the alignment of "Ours" and "Vanilla" with the ground truth across different label classes.

| Class | Ours Agreement (%) | Vanilla Agreement (%) |
|---|---|---|
| Hindu | 75.00 | 41.12 |
| Muslim | 95.00 | 100.00 |
| Neutral | 75.00 | 25.43 |
| Christian | 75.00 | 41.12 |

Table 7: Agreement of models with "hand" labels.

**Misclassification Analysis:**

The misclassification rates, i.e., the percentage of times each class was misclassified, were computed and are presented in Table 8. Notably, "Ours" performs consistently across all classes, whereas "Vanilla" struggles particularly with the "Neutral" class.

| Class | Ours Misclassification (%) | Vanilla Misclassification (%) |
|---|---|---|
| Hindu | 25.00 | 58.88 |
| Muslim | 5.00 | 0.00 |
| Neutral | 20.96 | 74.57 |
| Christian | 25.00 | 58.88 |

Table 8: Misclassification rates by class.

**Confusion Matrix Details:**

To further analyze the misclassifications, we present the confusion matrices for "Ours" and "Vanilla". These matrices illustrate the specific classes that each true class was misclassified as, providing insights into the model's weaknesses and guiding further improvements.

| True Class | Christian | Hindu | Muslim | Neutral |
|---|---|---|---|---|
| Christian | 26 | 0 | 0 | 16 |
| Hindu | 1 | 50 | 1 | 39 |
| Muslim | 1 | 0 | 57 | 2 |
| Neutral | 36 | 9 | 16 | 230 |

Table 9: Confusion Matrix for "Ours" model.

| True Class | Christian | Hindu | Muslim | Neutral |
|---|---|---|---|---|
| Christian | 32 | 3 | 1 | 6 |
| Hindu | 4 | 65 | 21 | 1 |
| Muslim | 0 | 0 | 60 | 0 |
| Neutral | 110 | 45 | 62 | 74 |

Table 10: Confusion Matrix for "Vanilla" model.

From these matrices, it is evident that both models show varying degrees of misclassification. The "Ours" model frequently confuses "Neutral" with "Christian," while the "Vanilla" model shows significant confusion between "Neutral" and all other classes, particularly "Christian" and "Hindu." These findings highlight areas where model improvements can be targeted, especially in differentiating between similar classes.

