# OpenReview forum: "FairCoT: Enhancing Fairness in Diffusion Models via Chain of Thought Reasoning of Multimodal Language Models"
_ICLR.cc/2025/Conference — Submitted to ICLR 2025_

### Official Review · Reviewer_sNLe · 2024-11-03

**Soundness:** 2
**Presentation:** 2
**Contribution:** 2
**Rating:** 5
**Confidence:** 4

**Summary:**

This paper presents FairCoT, a framework designed to enhance fairness in text2image models by leveraging CoT. It includes iterative CoT refinement, attire-based attribute prediction, and bias assessment via entropy scores. Their experiments show that FairCoT achieves high fairness and diversity metrics without compromising image quality or relevance.

**Strengths:**

1. It tackles significant questions in AI.

2. They successfully adapt CoT to the field of Fairness and debiasing

3. this is a simple, model-agnostic method that can be applied to both open-source and closed-source models without requiring retraining or parameter adjustments.

4. They achieve good debiasing without lowering the quality of the generated image

**Weaknesses:**

1. While I generally enjoyed reading this paper, the excessive use of bullet points makes it appear less professional and somewhat like an unfinished outline.
2. The paper relies heavily on CLIP for attribute prediction, but since CLIP was not originally designed as an exhaustive demographic classifier, this reliance might lead to skewed results and miss more subtle and nuanced features.
3. I am also interested in understanding the typical length of the CoT path used in the experiments. Does it require a large number of inferences? Additionally, is CoT useful for other downstream applications in text-to-image models like image editing and classification?
4. In qualitative examples 5 and 6, I noticed that FairCoT often produces images with multiple people, while the baseline typically shows a single person. Does this mean that FairCoT sacrifices some control over the image? Additionally, in Figure 6, it appears that FairCoT introduces some green noise into the image.

**Questions:**

I would appreciate it if the authors addressed the questions mentioned in the weaknesses section, and I will raise the score if their responses are convincing.

---

> ### Author Response · Authors · 2024-11-23
> **Response to Reviewer sNLe**
>
> Thank you for your valuable feedback and the positive remarks about our paper. We have carefully considered your comments and made corresponding revisions to address the concerns raised.
>
> **W1: Use of Bullet Points**
>
> We appreciate this observation and have revised the manuscript to eliminate bullet lists, opting instead for a more narrative style that enhances the flow and professional appearance of the text. All updates are available in the modified manuscript (updated methodology on pages 4-6; and updated experimental results on pages 9 and 10 that we highlighted in red).
>
> **W2: Concern About Reliance on CLIP for Attribute Prediction**
>
> CLIP was used to accurately predict attributes in several related works [2,3], after showing impressive gender and racial attribute prediction performance in the original CLIP paper [1], providing an automated method to label attributes and being more reliable than classical classifiers due to its zero-shot capabilities. To ensure results are more accurate, we integrated wider demographic attributes for races that are hard to distinguish visually, such as combining White, Middle-Eastern, and Hispanic Latino, into one class and Southeast Asian and East Asian into one class, and considering two age groups young and old inspired by bias literature [4]. Furthermore, we have extended the application of CLIP to include attire attributes beyond just gender and race, enabling powerful automated labeling that applies to a broader range of tasks beyond demographic classification.
>
> **W3: Length of CoT Path**
>
> As detailed in the appendix, the average CoT path length in our experiments is approximately 3.6 steps, with a maximum of 6 and a minimum of 3 (section 6.2.2). Regarding its applicability for other applications, while our current expertise is primarily in fairness for vision language models, our methodology can potentially be adapted for other vision-language tasks such as zero-shot image classification, Visual Question Answering (VQA), and image captioning with minor modification related to the desired tasks.
>
> **W4: Qualitative Examples and Image Control**
>
> The images including several people were indeed intentional, aiming to demonstrate the method’s capability to generate diverse scenarios,  and were limited to professions that interact with humans (e.g. a doctor in various environments that has to interact with patients). We clarified this in the updated appendix and provided additional comparative results with other baselines while including our face-shot results (we generated both face-shots and full body images), and other tasks like Judge image generation. Regarding the green noise observed in some images, this is attributed to the specific version of Stable Diffusion used SDXL for the doctor generation task. This artifact varies between versions and is a known issue with the model rather than a direct consequence of our FairCoT approach that does not alter the model weights. Therefore, we included further qualitative results for other tasks, baselines, and models in the appendix (Figures 6,7,8,9 on pages 14-17).
>
>
> [1] Shen, X., Du, C., Pang, T., Lin, M., Wong, Y., & Kankanhalli, M. (2024). Finetuning text-to-image diffusion models for fairness. In The Twelfth International Conference on Learning Representations.
>
> [2]Shrestha, R., Zou, Y., Chen, Q., Li, Z., Xie, Y., & Deng, S. (2024). FairRAG: Fair human generation via fair retrieval augmentation. In Proceedings of the IEEE/CVF Conference on Computer Vision and Pattern Recognition (pp. 11996-12005).
>
> [3]Chuang, C. Y., Jampani, V., Li, Y., Torralba, A., & Jegelka, S. (2023). Debiasing vision-language models via biased prompts. arXiv preprint arXiv:2302.00070.
>
> [4]Shen, X., Du, C., Pang, T., Lin, M., Wong, Y., & Kankanhalli, M. (2024). Finetuning text-to-image diffusion models for fairness. In The Twelfth International Conference on Learning Representations.

---

> > ### Comment · Reviewer_sNLe · 2024-11-24
> >
> > Thank you for the prompt response and for polishing the manuscript. Overall, I agree with the other reviewers that the contribution and technical innovation are somewhat limited, and the experiment may exhibit some bias. However, I believe the manuscript still provides interesting insights and demonstrates a commendable attempt at practical application. On balance, I will maintain my score.

---

> ### Author Response · Authors · 2024-11-30
> **Response to Reviewer sNLe(1/2)**
>
> Thank you for your prompt response and for recognizing the practical insights our manuscript provides. We appreciate your acknowledgment of our efforts to tackle significant questions in AI and your constructive feedback, which has been invaluable in refining our work.
>
> **Contribution and Technical Innovation**
>
> We understand your concern regarding the perceived limitations in our contribution and technical innovation. Our contributions are multifaceted and lie primarily in the novel adaptation of the Chain-of-Thought (CoT) technique to enhance fairness in text-to-image (T2I) generative models. While CoT has been employed in language models for reasoning tasks, to the best of our knowledge, its application to bias mitigation in image generation is unprecedented.
> Our method involves generating fair chains of thought using a Large Language Model (LLM) capable of generating images. We allow the model to iteratively reassess its reasoning until it produces fair images. These chains of thought are designed to generalize to any LLM that generates images or any text-to-image model connected to an LLM capable of reasoning. To enable adoption with any text-to-image model, we propose using an LLM to produce refined text prompts that can guide any Stable Diffusion (SD) model toward fairer outputs.
> Moreover, we extend our assessment of bias and fairness beyond classical types such as gender or race, aiming to make fairness research more inclusive of underrepresented minorities (e.g., women wearing hijab) who are often overlooked in the literature. To improve the zero-shot performance of CLIP in detecting people wearing religious attire, we proposed an attire-based method.
> Our method successfully debiased three versions of Stable Diffusion (SD), in addition to the DALL-E model, demonstrating its generality. It is model-agnostic and leverages LLMs to generate refined prompts without the need for fine-tuning or additional training data. Furthermore, our method generalized to multi-face generation and multi-concept scenarios beyond professions, such as images involving children, laptops, and animals.
>
>
> **Bias in the Experiment**
>
> We acknowledge your concern that the experiment may exhibit some bias.Our decision to use CLIP for attribute prediction was based on its robust zero-shot capabilities, ensuring accurate results even on images unseen in training data, such as low-quality images generated by text-to-image models or those involving underrepresented attributes. Classical classifiers often fail in such scenarios; for example, the DeepFace classifier incorrectly predicts a white female wearing a hijab as a Black male, regardless of the actual religion or attire.
>
> To cross-validate our results, we compared CLIP's attribute predictions with those of DeepFace [1-3] on 500 images generated by our method . We also compared these predictions against annotations from expert labelers on Amazon Mechanical Turk for age, gender, and race attributes. The results confirmed that CLIP provides more reliable attribute predictions in this context.
>
> |    Agreement with hand labels (%)     | **Gender** | **Race** | **Age** |
> |----------|------------|----------|---------|
> | **CLIP**     | 78         | 70       | 91      |
> | **DeepFace** | 52         | 50       | 84      |
>
> For attributes like religion, we introduced an attire-based enhancement to improve CLIP's attribute classification accuracy. This method reduces reliance on potentially biased features by focusing on neutral, observable attributes associated with specific religions.
>
> We recognize that biases in CLIP might have led to lower CLIP-T scores for alignment due to its inherent biases (e.g., predicting a female doctor as a nurse, as discussed in the literature). To mitigate this, we added a penalty during the training of our chain of thought for the CLIP-T score, interrupting training if the CLIP-T score falls below a threshold (alpha × CLIP-T0) where alpha<1. Notably, the alignment score at inference achieved comparable performance regardless of CLIP's biases.
> Our CoT refinement process is designed to counteract biases by encouraging the LLM to reason toward diverse and representative image generation. By iteratively refining chains of thought, we aim to balance the representation of different demographic groups in the generated images.
>
> Our qualitative and quantitative results demonstrate that our approach significantly reduces bias compared to baseline models, even when considering the inherent biases of CLIP and MLLMs.

---

> ### Author Response · Authors · 2024-11-30
> **Response to Reviewer sNLe(2/2)**
>
> We also utilized Llama-3.2-11B-Vision-Instruct for inference to validate the efficiency of our method. A sample of the prompts is provided in the appendix on page 25 sec 6.3.5 of the updated manuscript.
> | **Experiment**          | **Model**     | **Gender** | **Race** | **Age** | **Religion** | **CLIP-T** |
> |-------------------------|---------------|------------|----------|---------|--------------|------------|
> | Experimental Results                        | sdv1-5        | 0.92       | 0.85     | 0.43    | 0.74         | 0.25       |
> |                         | sdxl-turbo    | 0.93       | 0.77     | 0.41    | 0.77         | 0.26       |
> |                         | sd v2-1       | 0.92       | 0.86     | 0.39    | 0.72         | 0.25       |
> |  Multi-Face                       | sdv1-5        | 0.87       | 0.86     | 0.44    | 0.76         | 0.25       |
> |                         | sdxl-turbo    | 0.80       | 0.80     | 0.47    | 0.60         | 0.25       |
> |                         | sd v2-1       | 0.88       | 0.84     | 0.52    | 0.74         | 0.25       |
> |   Multi-Concept                      | sdv1-5        | 0.97       | 0.95     | 0.69    | 0.72         | 0.27       |
> |                         | sdxl-turbo    | 0.90       | 0.84     | 0.58    | 0.68         | 0.27       |
> |                         | sd v2-1       | 0.95       | 0.84     | 0.74    | 0.63         | 0.27       |
>
> The performance of our method using llama-3.2-11B-Vision-Instruct was comparable to that using GPT-3.5 Turbo. We observed similar improvements in fairness and diversity metrics across the evaluated attributes compared to other baselines, and further improvements in age attribute compared to GPT-3.5 FairCoT.
> Moreover, LLaMA achieved state-of-the-art performance in multi-concept generation, revealing its generalization power compared to GPT-3.5. Interestingly, this demonstrates the power of the vision modality in improving the reasoning capabilities of LLMs, as earlier versions (e.g., LlaMA 3.1) fail and hallucinate when prompted to perform this task.
>
> References
>
> [1] Serengil, Sefik, and Alper Ozpinar. "A Benchmark of Facial Recognition Pipelines and Co-Usability Performances of Modules." Journal of Information Technologies, vol. 17, no. 2, 2024, pp. 95–107, doi:10.17671/gazibtd.1399077. Accessed at https://dergipark.org.tr/en/pub/gazibtd/issue/84331/1399077.
>
> [2] Serengil, Sefik Ilkin, and Alper Ozpinar. "LightFace: A Hybrid Deep Face Recognition Framework." 2020 Innovations in Intelligent Systems and Applications Conference (ASYU), IEEE, 2020, pp. 23–27, doi:10.1109/ASYU50717.2020.9259802. Accessed at https://ieeexplore.ieee.org/document/9259802.
>
> [3] Serengil, Sefik Ilkin, and Alper Ozpinar. "HyperExtended LightFace: A Facial Attribute Analysis Framework." 2021 International Conference on Engineering and Emerging Technologies (ICEET), IEEE, 2021, pp. 1–4, doi:10.1109/ICEET53442.2021.9659697. Accessed at https://ieeexplore.ieee.org/document/9659697.

---

### Official Review · Reviewer_PNnP · 2024-11-03

**Soundness:** 1
**Presentation:** 3
**Contribution:** 3
**Rating:** 3
**Confidence:** 4

**Summary:**

The present paper proposes a novel method to tackle the lack of diversity in text-to-image models’ outputs displaying persons. To this end, the authors make use of pre-trained language models and CLIP and a discovery stage employing a Chain-of-thought process to create a set of prompt extensions. The approach is evaluated based on a prompt set covering gender, race, age, and religion and compared to a wide range of existing "debiasing" approaches.

**Strengths:**

- The paper addresses a current and crucial problem in research and industry as exemplified by text-to-image model applications such as DALL-E and Gemini being restricted in their capabilities of generating humans by their creators to prevent the display of social biases and, in turn, a negative social impact.

- While simple, the introduced method is flexible and model-agnostic.

- The paper provides an extensive overview of related work.

- Extensive evaluation covering multiple diffusion models and other „debiasing“ approaches.

- The evaluation goes beyond the display of single persons and single concepts commonly used in previous work.

**Weaknesses:**

- Instead of leveraging existing benchmark datasets such as [1], the evaluation is based on prompts selected by the authors.
Using existing datasets would enhance the comparability and reproducibility of results.

- More importantly, my main concern is the use of CLIP-T within both the methodology and evaluation metric, which could lead to a flawed evaluation. Therefore, I would urge the redesign of the evaluation process.

- While the prompts used are listed within the Appendix, it would increase clarity if at least the number of prompts for each category is listed in the main text.

- Section 4.3 lacks details on the hand-labeled images. Can you provide details on who performed the labeling, the annotators’ qualifications, the evaluation criteria, and the process used? Further, will this dataset be shared as part of the contribution?

- Figure and table captions are very sparse. More descriptive captions would improve clarity.

References:

[1] Luccioni et al. (NeurIPS 2023): https://proceedings.neurips.cc/paper_files/paper/2023/file/b01153e7112b347d8ed54f317840d8af-Paper-Datasets_and_Benchmarks.pdf

**Questions:**

- In Section 3.1, the term “training stage” may be misleading, as no training is conducted. This phase appears to be a bias discovery stage rather than training. Using “training” could lead readers to believe that model parameters are being tuned.

- The appendix indicates the use of GPT-3.5 as the language model. To improve reproducibility, please specify the exact version used. Additionally, have other open-weight language models been tested to reduce reliance on proprietary models?

**Minor comments**
- The abbreviation “MLLM” is introduced multiple times.
- There are missing spaces in several locations, such as in line 266 “(Figure1)” and line 410 “fairness.Further.”

---

> ### Author Response · Authors · 2024-11-23
> **Response to Reviewer PNnP**
>
> Thank you for your thorough review and constructive feedback. Your suggestions have been instrumental in refining our methodology and presentation. Below, we detail how we have addressed each of the points raised.
>
> **W1: Evaluation Using Author-Selected Prompts**
>
> Our choice to use profession-based prompts aligns with methods utilized in baseline comparisons such as finetuning, debiasing VL, and fair Difussion methods, ensuring consistency across evaluations and highlighting our approach's state-of-the-art capabilities [1-4]. Our method also demonstrates generalizability beyond typical profession-based scenarios, accommodating diverse settings and subjects, as illustrated in sec.4.2.2, 4.2.3, 6.2.2. Moreover, the professions list was generated by an LLM to guarantee fair evaluation.
>
> **W2: Concern About CLIP-T Usage in Methodology and Evaluation**
>
> We employed CLIP-T to ensure that the generated images align closely with the initial prompts, addressing a common limitation where images deviate significantly from specified attributes (e.g., generating an image of a 'normal female' when 'female doctor' is intended), and this metric was widely used in the literature [1-2]. While CLIP-T primarily verifies prompt alignment, we evaluated fairness using normalized entropy.
>
> **W3: More Detailed Prompt Information in the Main Text**
>
> Acknowledging the feedback on prompt clarity, we have referenced to the list of prompts used directly in the main text (please refer to page 7 lines 345-347 in the updated manuscript). The omission was originally due to space constraints, but we understand the importance of transparency in prompt categorization to enhance reader comprehension.
>
> **W4: Details on Hand-Labeled Images**
>
> The labeling was performed by a PhD student using Amazon Turk Sandbox. We will include updated labels using Amazon Mechanical Turk, with quality control ensured by employing random master labelers. Random master labelers achieved 96.7% agreement with the initial labels provided by the PhD student. We are committed to sharing this hand-labeled dataset, along with our codes, upon acceptance of the paper to facilitate reproducibility and further research.
>
> **W4: Sparse Figure and Table Captions**
>
> We thank you for pointing out the sparsity of our captions. We have revised all figure and table captions to be more descriptive, thereby improving the clarity and informativeness of our visual and tabular presentations (please refer to the updated manuscript).
>
> **Q1: Misleading Term "Training Stage"**
>
> We intended this to refer to the training of the Chain of Thoughts (CoT), which does not involve further training post-generation. We revised this to the "CoT generation stage" to better reflect the process while aligning with the terminology used in CoT reasoning research (please refer to page 4 in the updated manuscript).
>
> **Q2: Specifics on Language Model Version and Alternatives**
>
> We specified that we used the 'GPT-3.5 Turbo' version during our initial studies on page 22. Our updated version in camera-ready will incorporate 'Llama-3.2-11B Vision Instruct' in inference testing to diversify our language model usage, enhancing the paper's reproducibility and reducing reliance on proprietary models. We plan to share the Llama-3.2-11B Vision Instruct results before the rebuttals deadline.
>
> **Q3: Editorial Corrections**
>
> The noted typographical errors, such as missing spaces, have been corrected. We appreciate your attention to detail.
>
> [1] Shen, X., Du, C., Pang, T., Lin, M., Wong, Y., & Kankanhalli, M. (2024). Finetuning text-to-image diffusion models for fairness. In The Twelfth International Conference on Learning Representations.
>
> [2]Shrestha, R., Zou, Y., Chen, Q., Li, Z., Xie, Y., & Deng, S. (2024). FairRAG: Fair human generation via fair retrieval augmentation. In Proceedings of the IEEE/CVF Conference on Computer Vision and Pattern Recognition (pp. 11996-12005).
>
> [3]Chuang, C. Y., Jampani, V., Li, Y., Torralba, A., & Jegelka, S. (2023). Debiasing vision-language models via biased prompts. arXiv preprint arXiv:2302.00070.
>
> [4]Friedrich, F., Brack, M., Struppek, L., Hintersdorf, D., Schramowski, P., Luccioni, S., & Kersting, K. (2023). Fair diffusion: Instructing text-to-image generation models on fairness. arXiv preprint arXiv:2302.10893.

---

> > ### Comment · Reviewer_PNnP · 2024-11-26
> >
> > Thank you for addressing my concerns and answering my questions.
> >
> > - W2: Similar to reviewer SnAZ, my concern remains that the biases present in CLIP and MLLM affect the evaluation and results of the proposed method.
> > - W4: Details such as evaluation criteria, and the process used are still not transparently provided. Could you provide the questionnaire used in the study? In 7.2 the authors state that a single human labeler is considered since the task is easy. However, this is not obvious to me given the varying quality of generated images.
> > - Q2: On page 22 the authors state that they "employ the OpenAI GPT-3.5 Turbo architecture for the LLM." There are multiple versions of gpt-3.5 turbo. Each model version is dated with either -MMDD or a YYYY-MM-DD suffix. See https://platform.openai.com/docs/deprecations. I'm looking forward to the Llama-3.2-11B Vision Instruct results.
> >
> > Currently, I would like to keep my score as it is.

---

> > > ### Author Response · Authors · 2024-11-30
> > > **Response to Reviewer PNnP(2/2)**
> > >
> > > **Q2: Clarification on GPT-3.5 Turbo Version and LLaMA Results**
> > >
> > > 1.	Specific GPT-3.5 Turbo Version Used
> > >
> > > We appreciate your attention to detail regarding the specific version of GPT-3.5 Turbo used in our experiments. We employed the "gpt-3.5-turbo-0613" version for our study, which was the latest stable release available at the time. We will update the manuscript to specify this exact version, enhancing the reproducibility of our work.
> > >
> > > 2.	Results with llama-3.2-11B-Vision-Instruct
> > >
> > > We have conducted additional experiments using llama-3.2-11B-Vision-Instruct to demonstrate the generalizability of our method with open-source models. In these experiments, we prompted LLaMA to generate a chain of thought inspired by those generated during the training phase for the inference task. We then requested it to generate prompts suitable for diffusion models based on this chain of thought. A sample of the prompts is provided in the appendix on page 25 sec 6.3.5 of the updated manuscript.
> > >
> > > | **Experiment**          | **Model**     | **Gender** | **Race** | **Age** | **Religion** | **CLIP-T** |
> > > |-------------------------|---------------|------------|----------|---------|--------------|------------|
> > > |  Experimental Results                       | sdv1-5        | 0.92       | 0.85     | 0.43    | 0.74         | 0.25       |
> > > |                         | sdxl-turbo    | 0.93       | 0.77     | 0.41    | 0.77         | 0.26       |
> > > |                         | sd v2-1       | 0.92       | 0.86     | 0.39    | 0.72         | 0.25       |
> > > |  Multi-Face                       | sdv1-5        | 0.87       | 0.86     | 0.44    | 0.76         | 0.25       |
> > > |                         | sdxl-turbo    | 0.80       | 0.80     | 0.47    | 0.60         | 0.25       |
> > > |                         | sd v2-1       | 0.88       | 0.84     | 0.52    | 0.74         | 0.25       |
> > > | Multi-Concept                        | sdv1-5        | 0.97       | 0.95     | 0.69    | 0.72         | 0.27       |
> > > |                         | sdxl-turbo    | 0.90       | 0.84     | 0.58    | 0.68         | 0.27       |
> > > |                         | sd v2-1       | 0.95       | 0.84     | 0.74    | 0.63         | 0.27       |
> > >
> > > The performance of our method using llama-3.2-11B-Vision-Instruct was comparable to that using GPT-3.5 Turbo. We observed similar improvements in fairness and diversity metrics across the evaluated attributes compared to other baselines, and further improvements in age attribute compared to GPT-3.5 FairCoT.
> > >
> > > Moreover, LLaMA achieved state-of-the-art performance in multi-concept generation, revealing its generalization power compared to GPT-3.5. Interestingly, this demonstrates the power of the vision modality in improving the reasoning capabilities of LLMs, as earlier versions (e.g., LlaMA 3.1) fail and hallucinate when prompted to perform this task.
> > > These additional experiments confirm that our method is generalizable and does not rely on proprietary models, enhancing the accessibility and reproducibility of our approach. We have included these results in the updated manuscript.
> > >
> > > References
> > >
> > > [1]Agarwal, Sandhini, et al. "Evaluating clip: towards characterization of broader capabilities and downstream implications." arXiv preprint arXiv:2108.02818 (2021).
> > >
> > > [2]Berg, Hugo, et al. "A prompt array keeps the bias away: Debiasing vision-language models with adversarial learning." arXiv preprint arXiv:2203.11933 (2022).
> > >
> > > [3] Serengil, Sefik, and Alper Ozpinar. "A Benchmark of Facial Recognition Pipelines and Co-Usability Performances of Modules." Journal of Information Technologies, vol. 17, no. 2, 2024, pp. 95–107, doi:10.17671/gazibtd.1399077. Accessed at https://dergipark.org.tr/en/pub/gazibtd/issue/84331/1399077.
> > >
> > > [4] Serengil, Sefik Ilkin, and Alper Ozpinar. "LightFace: A Hybrid Deep Face Recognition Framework." 2020 Innovations in Intelligent Systems and Applications Conference (ASYU), IEEE, 2020, pp. 23–27, doi:10.1109/ASYU50717.2020.9259802. Accessed at https://ieeexplore.ieee.org/document/9259802.
> > >
> > > [5] Serengil, Sefik Ilkin, and Alper Ozpinar. "HyperExtended LightFace: A Facial Attribute Analysis Framework." 2021 International Conference on Engineering and Emerging Technologies (ICEET), IEEE, 2021, pp. 1–4, doi:10.1109/ICEET53442.2021.9659697. Accessed at https://ieeexplore.ieee.org/document/9659697.

---

> ### Author Response · Authors · 2024-11-30
> **Response to Reviewer PNnP(1/2)**
>
> Thank you for your thoughtful follow-up and for giving us the opportunity to address your remaining concerns. We greatly appreciate your detailed feedback, which has helped us to further improve our work.
>
> **W2: Impact of Biases in CLIP and MLLMs on Evaluation and Results**
>
> We understand your concern regarding the potential propagation of biases present in CLIP and Multimodal Large Language Models (MLLMs) into our method and evaluation.
> We acknowledge that these models are not entirely free from biases, as highlighted in prior studies [1,2]. Our decision to use CLIP for attribute prediction was based on its robust zero-shot capabilities, ensuring accurate results even on images unseen in training data, such as low-quality images generated by text-to-image models or those involving underrepresented attributes. Classical classifiers often fail in such scenarios; for example, the DeepFace classifier incorrectly predicts a white female wearing a hijab as a Black male, regardless of the actual religion or attire.
> To cross-validate our results, we compared CLIP's attribute predictions with those of DeepFace [3-5] on 500 images generated by our method . We also compared these predictions against annotations from expert labelers on Amazon Mechanical Turk for age, gender, and race attributes. The results confirmed that CLIP provides more reliable attribute predictions in this context.
>
> |     Agreement with Hand Labels(%)     | **Gender** | **Race** | **Age** |
> |----------|------------|----------|---------|
> | **CLIP**     | 78         | 70       | 91      |
> | **DeepFace** | 52         | 50       | 84      |
>
> For attributes like religion, we introduced an attire-based enhancement to improve CLIP's attribute classification accuracy. This method reduces reliance on potentially biased features by focusing on neutral, observable attributes associated with specific religions.
>
> We recognize that biases in CLIP might have led to lower CLIP-T scores for alignment due to its inherent biases (e.g., predicting a female doctor as a nurse, as discussed in the literature). To mitigate this, we added a penalty during the training of our chain of thought for the CLIP-T score, interrupting training if the CLIP-T score falls below a threshold (alpha × CLIP-T0) where alpha<1. Notably, the alignment score at inference achieved comparable performance regardless of CLIP's biases.
> Our CoT refinement process is designed to counteract biases by encouraging the LLM to reason toward diverse and representative image generation. By iteratively refining chains of thought, we aim to balance the representation of different demographic groups in the generated images.
> Our qualitative and quantitative results demonstrate that our approach significantly reduces bias compared to baseline models, even when considering the inherent biases of CLIP and MLLMs.
>
> **W4: Transparency of Evaluation Criteria and Process**
>
> We appreciate your request for more transparency regarding our evaluation criteria and process. Below, we provide the details of the questionnaire used in our study:
>
> Questionnaire:
>
> •	Instruction: "Select the religion of the person based on the visible attire."
>
> •	Options:
>
> o	Neutral
>
> o	Muslim
>
> o	Christian
>
> o	Hindu
>
> o	Sikh
>
> o	Jewish
>
> The initial labeling was performed by a PhD student with expertise in fairness. To ensure the reliability of the labels, we conducted a validation study using Amazon Mechanical Turk (AMT) with Master Workers who had an approval rate of over 98% and more than 1,000 approved HITs. The agreement between the initial labels and those from AMT workers was 96.7%, indicating a high level of consistency.
>
> We considered the task manageable for a single labeler because the determination of religion was based primarily on visible attire, such as specific religious garments or symbols (e.g., hijab, cross), which are generally easy to identify regardless of the image quality. In cases where religious attire was not visible, the image was labeled as "Neutral."
> We recognize that the varying quality of generated images could potentially affect label accuracy. However, the high agreement rate between the initial labels and the AMT labels suggests that the labeling task was indeed straightforward and reliable.

---

### Official Review · Reviewer_SnAZ · 2024-11-04

**Soundness:** 2
**Presentation:** 2
**Contribution:** 1
**Rating:** 5
**Confidence:** 3

**Summary:**

The paper present FairCoT: an iterative CoT refinement and attire-based attribute prediction technique mitigate biases present in text to image generative models. The paper develops a MLLM and CLIP guided method to develop a CoT pool, which is then used to guide a T2I model towards generating de-biased outputs. Experiments on DALL-E and various Stable Diffusion variants demonstrate that FairCoT significantly improves fairness and diversity metrics without compromising image quality or relevance.

**Strengths:**

1. The paper tackles an important problem in T2I models - and aims to develop a method that makes them generate images which represent a broader sample of the society.
2. The paper studies multiple aspects such as age, religion, gender and race.

**Weaknesses:**

1. Technically limited contribution. The work iteratively applies CoT to generate multiple images to steer them towards being fair, which in my opinion does not offer many novel insights or findings.
2. The paper used MLLMS and CLIP throughout their pipeline, as a means to alleviate bias in T2I models. However, the paper does not account for the bias present in these models, which has been extensively studied in [1,2].
3. Computationally inefficient. The pipeline involves multiple calls to a T2I model. The authors could refer to [3] for a survey of methods on this topic which perform for example, efficient, embedding-based methods.

[1] https://arxiv.org/abs/2108.02818
[2] https://arxiv.org/abs/2203.11933
[3] https://arxiv.org/pdf/2404.01030

**Questions:**

1. In Section 3.1.2, why is "attire" chosen to be a good representation of a given religion? I am not sure about Figure 2? How does a man in a suit define the religion of that person? A person of a given religion might not always be wear a given attire. I don't think it is appropriate to generalize this.
2. I would like to understand how the pipeline works with the SD set of models, given they cannot be prompted the same way DALL-E can be since its integrated with GPT-4.
3. General suggestions - I think all the captions of the figures and tables should be improved.

---

> ### Author Response · Authors · 2024-11-23
> **Response to Reviewer SnAZ**
>
> Thank you for your thoughtful analysis and feedback. We have addressed each of your concerns as outlined below, aiming to clarify our contributions and methodologies.
>
> **W1: Technically Limited Contribution**
>
> Our key contribution is adapting CoT effectively to enhance fairness in image generation, which significantly advances over existing methods where traditional automatic CoT adjustments fail [1]. This contribution is both state-of-the-art and practical, requiring no additional training while being generalizable across various tasks. Furthermore, our attire-based CLIP enhancement method improves zero-shot classification performance in CLIP for religion, which can be extended beyond addressing bias attributes prediction.
>
> **W2: Biases in MLLMs and CLIP**
>
> CLIP is specifically utilized for attribute prediction due to its efficiency and accuracy in predicting gender, race, and age [2], as recognized in the relevant bias literature [3,4]. We also proposed our attire-based method to enhance CLIP's attribute classification for complex attributes like religion(sec 4.3). On the other hand, our chain of thought refinement method aims to debias MLLMs for image generation task by stimulating the reasoning of MLLMs for fairness resulting in more equitable image generation. We measure bias using normalized entropy, with both qualitative and quantitative results demonstrating the fairness of our images across various demographic attributes (We included additional qualitative results in the appendix of the updated manuscript).
>
> **W3: Computational Efficiency**
>
> In practice, our pipeline is optimized for efficiency; after initially training seven CoTs, no further training is required. During inference, for any given task, only a single call per prompt to Stable Diffusion is necessary, as detailed in Figure 1. This method avoids the need for finetuning, RAG, external datasets, or additional GPUs, making it both effective and practical.
>
> **Q1:Choice of "Attire" as a Representation of Religion**
>
> We did not intend to imply that attire directly signifies religion; rather, our goal was to enhance fairness by representing individuals in diverse attires who are often underrepresented. This point is clarified further in our report and noted as a limitation (please refer to the updated manuscript on page 10 lines 514-519).
>
> **Q2: Compatibility with the SD Set of Models**
>
> As depicted in Figure 1, an LLM generates a set of SD-compatible prompts from the CoT, which are then used to generate images. This method not only adds reasoning capabilities to SD models but also enhances their performance in addressing bias and fairness (please refer to sec.6.3.3-6.3.5 in the appendix of the updated manuscript on pages 23-26).
>
> **Q3: Improvement of Figure and Table Captions**
>
> Thank you for suggesting improvements to the captions of our figures and tables. We revised all captions to enhance clarity and relevance (please refer to the updated manuscript for improved captions).
>
> [1]Shaikh, O., Zhang, H., Held, W., Bernstein, M., & Yang, D. (2023). On second thought, let’s not think step by step! Bias and toxicity in zero-shot reasoning. In A. Rogers, J. Boyd-Graber, & N. Okazaki (Eds.), Proceedings of the 61st Annual Meeting of the Association for Computational Linguistics (Volume 1: Long Papers) (pp. 4454–4470). Association for Computational Linguistics. https://doi.org/10.18653/v1/2023.acl-long.244
>
> [2]Radford, A., Kim, J. W., Hallacy, C., Ramesh, A., Goh, G., Agarwal, S., ... & Sutskever, I. (2021, July). Learning transferable visual models from natural language supervision. In International conference on machine learning (pp. 8748-8763). PMLR.
>
> [3]Shrestha, R., Zou, Y., Chen, Q., Li, Z., Xie, Y., & Deng, S. (2024). FairRAG: Fair human generation via fair retrieval augmentation. In Proceedings of the IEEE/CVF Conference on Computer Vision and Pattern Recognition (pp. 11996-12005).
>
> [4]Chuang, C. Y., Jampani, V., Li, Y., Torralba, A., & Jegelka, S. (2023). Debiasing vision-language models via biased prompts. arXiv preprint arXiv:2302.00070.

---

> > ### Comment · Reviewer_SnAZ · 2024-11-24
> > **Thanks for your rebuttal!**
> >
> > I appreciate the rebuttal made by the authors.
> >
> > Regarding the weaknesses :
> >
> > 1. I still am of the opinion that the contributions are limited. I am also not sure how well the CoT technique generalizes to other models.
> > 2. My question was more around how much do the biases present in CLIP and MLLM affect the results of this method.
> > 3. Agreed, the model does not need fine-tuning or RAG, but multiple inference calls is costly as well.
> >
> > Thank you for answering my questions. I am not yet clear on how SD fits into the Figure 3 pipeline.
> >
> > Overall, I would like to keep my score.

---

> ### Author Response · Authors · 2024-11-30
> **Response to Reviewer SnAZ (1/3)**
>
> Thank you for your continued engagement and for providing further feedback on our work. We value your insights and would like to address your remaining concerns to clarify our contributions and methodologies.
>
> **Limited Contributions and Generalization of the CoT Technique**
>
> We understand your concern regarding the novelty and generalization of our approach. Our contributions are multifaceted and lie primarily in the novel adaptation of the Chain-of-Thought (CoT) technique to enhance fairness in text-to-image (T2I) generative models. While CoT has been employed in language models for reasoning tasks, to the best of our knowledge, its application to bias mitigation in image generation is unprecedented.
>
> Our method involves generating fair chains of thought using a Large Language Model (LLM) capable of generating images. We allow the model to iteratively reassess its reasoning until it produces fair images. These chains of thought are designed to generalize to any LLM that generates images or any text-to-image model connected to an LLM capable of reasoning. To enable adoption with any text-to-image model, we propose using an LLM to produce refined text prompts that can guide any Stable Diffusion (SD) model toward fairer outputs.
>
> Moreover, we extend our assessment of bias and fairness beyond classical types such as gender or race, aiming to make fairness research more inclusive of underrepresented minorities (e.g., women wearing hijab) who are often overlooked in the literature. To improve the zero-shot performance of CLIP in detecting people wearing religious attire, we proposed an attire-based method.
> Our method successfully debiased three versions of Stable Diffusion (SD), in addition to the DALL-E model, demonstrating its generality. It is model-agnostic and leverages LLMs to generate refined prompts without the need for fine-tuning or additional training data. Furthermore, our method generalized to multi-face generation and multi-concept scenarios beyond professions, such as images involving children, laptops, and animals.
>
> We conducted experiments using Llama-3.2-11B-Vision-Instruct for inference and reported the results in the updated manuscript. A sample of the prompts generated by LLaMA is provided in the appendix sec. 6.3.5 on page 25. We also included the multi-face and multi-concept results on page 8. Below are the complete results:
>
> | **Experiment**          | **Model**     | **Gender** | **Race** | **Age** | **Religion** | **CLIP-T** |
> |--------------|---------------|------------|----------|---------|--------------|------------|
> |  Experimental Results   | sdv1-5        | 0.92       | 0.85     | 0.43    | 0.74         | 0.25       |
> |                         | sdxl-turbo    | 0.93       | 0.77     | 0.41    | 0.77         | 0.26       |
> |                         | sd v2-1       | 0.92       | 0.86     | 0.39    | 0.72         | 0.25       |
> |    Multi-Face     | sdv1-5        | 0.87       | 0.86     | 0.44    | 0.76         | 0.25       |
> |                         | sdxl-turbo    | 0.80       | 0.80     | 0.47    | 0.60         | 0.25       |
> |                         | sd v2-1       | 0.88       | 0.84     | 0.52    | 0.74         | 0.25       |
> |   Multi-Concept     | sdv1-5        | 0.97       | 0.95     | 0.69    | 0.72         | 0.27       |
> |                         | sdxl-turbo    | 0.90       | 0.84     | 0.58    | 0.68         | 0.27       |
> |                         | sd v2-1       | 0.95       | 0.84     | 0.74    | 0.63         | 0.27       |
>
> Additionally, our method offers several advantages over existing prompt-based and fine-tuning methods. Prompt-based methods require human labor, may introduce bias, perform minor changes, and can lead to inadvertent consequences. Fine-tuning methods are computationally expensive, can lead to memorization, and do not generalize to other bias types not included in the fine-tuning process.

---

> ### Author Response · Authors · 2024-11-30
> **Response to Reviewer SnAZ (2/3)**
>
> **Impact of Biases in CLIP and MLLMs on Our Method**
>
> We appreciate your concern regarding the potential biases present in CLIP and Multimodal Large Language Models (MLLMs) and their effect on our method's results.
> We acknowledge that these models are not entirely free from biases, as highlighted in prior studies [1,2]. Our decision to use CLIP for attribute prediction was based on its strong zero-shot capabilities, which help ensure accurate results even on images that are unseen during training, such as low-quality images generated by text-to-image models or those involving underrepresented attributes. Classical classifiers often fail to generalize well in such scenarios. For instance, the DeepFace classifier [3-5] incorrectly predicts a white female wearing a hijab as a Black male, regardless of the actual religion or attire.
> To validate our results, we compared CLIP's attribute predictions with those of DeepFace on 500 images generated by our method. We also compared these predictions against annotations from expert labelers on Amazon Mechanical Turk for age, gender, and race attributes. The findings confirmed that CLIP provides more reliable attribute predictions in this context.
>
> |     Agreement with Hand labels(%)     | **Gender** | **Race** | **Age** |
> |----------|------------|----------|---------|
> | **CLIP**     | 78         | 70       | 91      |
> | **DeepFace** | 52         | 50       | 84      |
>
> For attributes like religion, we introduced an attire-based enhancement to improve CLIP's attribute classification accuracy. This approach reduces reliance on potentially biased features by focusing on neutral, observable attributes associated with specific religions.
>
> We recognize that biases in CLIP might have led to lower CLIP-T scores for alignment due to its inherent biases (e.g., predicting a female doctor as a nurse, as discussed in the literature). To mitigate this, we added a penalty during the training of our chain of thought for the CLIP-T score, interrupting training if the CLIP-T score falls below a threshold (alpha × CLIP-T0) where alpha<1. Notably, the alignment score at inference achieved comparable performance regardless of CLIP's biases.
>
> Our CoT refinement process is designed to counteract biases by encouraging the LLM to reason toward diverse and representative image generation. By iteratively refining chains of thought, we aim to balance the representation of different demographic groups in the generated images.
>
> Our qualitative and quantitative results demonstrate that our approach significantly reduces bias compared to baseline models, even when considering the inherent biases of CLIP and MLLMs.
>
> **Computational Cost of Multiple Inference Calls**
>
> We understand your concern regarding the computational cost associated with multiple inference calls in our method. While our pipeline does involve additional computations compared to a single-pass approach, we have taken steps to optimize efficiency and manage the computational overhead.
>
> Firstly, the initial generation of the chain-of-thought (CoT) pool is performed offline. Once generated, the CoT pool can be reused across different prompts and tasks, eliminating the need for repeated CoT computations during inference.
> Secondly, during inference, each input prompt requires only one additional LLM call to generate the refined prompt, followed by a single text-to-image model inference. This approach minimizes the number of additional inference calls while achieving significant performance gains in fairness.
>
> We conducted a cost-benefit analysis, including runtime and resource utilization, which we have included in sec. 6.2.2 on page 22 of the updated manuscript. The analysis demonstrates that the computational overhead is manageable and justified by the substantial improvements in fairness and diversity metrics achieved by our method.
>
> Moreover, we believe that the additional computational cost is acceptable for fair image generation that generalizes to any bias type or concept, especially since it can be applied upon request—when fair generation is specifically needed by the user.

---

> ### Author Response · Authors · 2024-11-30
> **Response to Reviewer SnAZ (3/3)**
>
> **Clarification on How SD Fits into the Figure 3 Pipeline**
>
> We apologize for any confusion regarding the integration of Stable Diffusion (SD) into our pipeline. Allow us to provide a clearer explanation:
>
> Figure 3 Pipeline: In our pipeline depicted in Figure 3, the MLLM generates a refined Chain-of-Thought (CoT) process for fair image generation.
>
> Enhancing SD with MLLM Reasoning: At inference (Figure1) the MLLM generate prompts the are specifically formatted to be compatible with SD models.By incorporating MLLM-generated reasoning into the prompts, we provide SD models with enriched context, leading to fairer image generation without modifying the SD models themselves.
>
> Instead of using handcrafted prompts written by humans to generate fairer images with SD, we utilize prompts generated by an MLLM. These MLLM-generated prompts are designed to be fair and specifically targeted for image generation, as they are produced by an MLLM capable of generating images. To adapt our approach to SD, we generate prompts that effectively guide the SD model's image generation process.
>
> Similar to research in reasoning where prompting LLMs to "think step by step" improves their reasoning compared to hand-crafted chains of thought, we demonstrate that MLLM-generated chains of thought enhance fairness in image generation more effectively than human prompting and fine-tuning methods.
>
> Our approach equips SD models with improved context derived from the MLLM's reasoning process, enabling them to generate images that are fairer and more representative without altering the SD models themselves. This integration ensures that the benefits of the MLLM's reasoning are seamlessly transferred to the SD models through the refined prompts.
>
> References
>
> [1]Agarwal, Sandhini, et al. "Evaluating clip: towards characterization of broader capabilities and downstream implications." arXiv preprint arXiv:2108.02818 (2021).
>
> [2]Berg, Hugo, et al. "A prompt array keeps the bias away: Debiasing vision-language models with adversarial learning." arXiv preprint arXiv:2203.11933 (2022).
>
> [3] Serengil, Sefik, and Alper Ozpinar. "A Benchmark of Facial Recognition Pipelines and Co-Usability Performances of Modules." Journal of Information Technologies, vol. 17, no. 2, 2024, pp. 95–107, doi:10.17671/gazibtd.1399077. Accessed at https://dergipark.org.tr/en/pub/gazibtd/issue/84331/1399077.
>
> [4] Serengil, Sefik Ilkin, and Alper Ozpinar. "LightFace: A Hybrid Deep Face Recognition Framework." 2020 Innovations in Intelligent Systems and Applications Conference (ASYU), IEEE, 2020, pp. 23–27, doi:10.1109/ASYU50717.2020.9259802. Accessed at https://ieeexplore.ieee.org/document/9259802.
>
> [5] Serengil, Sefik Ilkin, and Alper Ozpinar. "HyperExtended LightFace: A Facial Attribute Analysis Framework." 2021 International Conference on Engineering and Emerging Technologies (ICEET), IEEE, 2021, pp. 1–4, doi:10.1109/ICEET53442.2021.9659697. Accessed at https://ieeexplore.ieee.org/document/9659697.

---

> > ### Comment · Reviewer_SnAZ · 2024-12-03
> > **Thanks again!**
> >
> > I appreciate the efforts.  Here are some follow-up questions/suggestions :
> >
> > 1. My point about the inherent bias in CLIP and MLLMs is not mitigated by the additional comparison with DeepFake. Accuracy is not a good measure in this case. A better experiment would be to show that even if there is a bias in CLIP and MLLM, FairCoT is able to overcome it.
> >
> > 2. What does enriched context in case of SD models mean? Are they more descriptive? If yes, most SD models (1.5 and 2.1) have a 77 token limit so enriched descriptions might not always be helpful.
> >
> > I am increasing the score to acknowledge the efforts and the (new) experiments made by the authors, however I still believe the core contribution is limited.

---

> ### Author Response · Authors · 2024-12-03
> **Thank you for your efforts!**
>
> Thank you so much for all your feedback and help! We will take these into consideration later in our manuscript.
>
> 2. The enriched context is still within the tokens limit of SD. In fact, it was one of the inspirations for using an LLM to generate prompts that are SD-compatible. We previously tried using a short summary of the original CoT and feeding it to the SD model. Below are the SD-XL results for using summary CoT:
> | Model                | Prompt       | Gender | Race  | Age   | Religion | CLIP-T↑ |
> |----------------------|--------------|--------|-------|-------|----------|---------|
> | **SDXL**  | summary CoT     | 0.80   | 0.51  | 0.85  | 0.66     | 0.254  |

---

### Official Review · Reviewer_wptm · 2024-11-05

**Soundness:** 2
**Presentation:** 2
**Contribution:** 2
**Rating:** 3
**Confidence:** 4

**Summary:**

This paper presents FairCoT, a new framework for improving fairness in text-to-image diffusion models by leveraging Chain-of-Thought (CoT) reasoning within multimodal large language models (MLLMs). The key idea is to use iterative CoT refinement and attire-based attribute prediction to systematically reduce biases in generated images. Experiments across multiple models show improved fairness and diversity without sacrificing image quality or relevance.

**Strengths:**

- Reducing the biases and improving fairness in image generation models are very important topics for both research communities and the real-world applications of the models.
- The proposed approach shows significant improvement over gender, race, age, and religion attributes while maintaining image quality (measured via CLIP-T scores).

**Weaknesses:**

- The technical contributions and the novelty of this work are not significant. Previous works [1] also explored using an iterative strategy to improve the fairness of Text-to-Image diffusion models.
- In addition, the authors did not compare the several more relevant baselines, such as IDA [1].
- It is not clear how the approach iterative updates the CoT and consequently improves the fairness of the images. It would be helpful if the authors could provide a more clear and concrete example of the refinement process.
- The presentation of the paper is not clear and should be further improved. For example, the diagram shown in Figure 3 is not very clear.

[1] Debiasing Text-to-Image Diffusion Models. He, et al. 2024.

**Questions:**

- While the quantitative results of CLIP-T score only have small variations, in the qualitative comparison in Figure 6, the images generated by FairCoT look pretty weird and of poor quality. Could you have an explanation for this? Is this a common issue of FairCoT?

---

> ### Author Response · Authors · 2024-11-23
> **Response to Reviewer wptm**
>
> Thank you for your insightful comments and suggestions. We value your feedback and have carefully considered each point to enhance the clarity and impact of our work.
>
> **W1:  Technical Contribution is limited and [1] used iterations for fairness**
>
> We appreciate the comparison to prior work by He et al. [1], which we were not aware of as it was published on October 28, 2024 after the ICLR submission deadline. We have referred to it in the updated version as concurrent work. Here, we highlight the important differences.
> First, the learning phase of our approach constructs several templates for Chain of Thought (CoT), that is, improving the MLLM reasoning for fair image generation that will be used to extract appropriate prompts that promote fairness. In contrast to [1], which fine-tunes the model by updating its weights, our approach does not require access to the model parameters and therefore is applicable to closed-source models.
> Second, at inference stage, our model selects a chain of thought that is closer to CoTs that are constructed at the learning stage and is shown to generalize to multiple biases on which it hasn’t seen at the learning stage -- in sec. 4.2.2, 4.2.3, 6.2.2 we have shown that for multi-face generation, multi-concept generation, including laptops, kids, and dog breeds. This is in contrast to [1] which requires retraining for new types of biases and has been evaluated only in the case of gender and ethnicity bias with primary experiments on 6 professions.
> We expanded our literature review to clarify these distinctions and highlight the novel aspects of our approach, as highlighted in the updated version lines 115-119.
>
> **W2: Comparison with IDA as Relevant Baseline**
>
> IDA was published after our submission and has yet to make its model weights or code public.
> For this reason, we are unable to obtain a direct comparison on all the biases that we address and on the metrics that we used including normalized entropy and CLIP-T score (and used in the literature [2,3]).
> However, since the paper reports results in terms of gender KL divergence over 6 professions, we evaluated our gender KL divergence on 2 common professions we share: Housekeeper and Teacher where we achieved superior performance of 0 divergence between genders. We will include those results in the updated manuscript. We commit to including IDA in our baseline comparisons and conducting a thorough evaluation once their resources become available.
>
> **W3: Clarification of Iterative CoT Updates**
>
> To better illustrate our method, we have included samples of how the chain of thought improves from zero-shot CoT, to the first iteration, and later to convergence (sec. 6.3.1-6.3.3 in the updated version). We also added for inference, an example of doctor chain of thought generation and text prompts generation to show how the trained Cots help in debiasing other tasks. This process exemplifies how our method can be generalized to other contexts, such as generating fair images for different medical professions (sec. 6.3.4 and 6.3.5  in the updated version). We will include a detailed toy example in the appendix of the final version. This example will walk through the iterative refinement process using nurse images, showing how feedback on bias and alignment is used to adjust CoTs until optimal fairness metrics are achieved.
>
> **W4: Presentation Improvements**
>
> We have revised Figure 3 and related visual materials to enhance clarity (please refer to updated Figure 3 on Page 6). Moreover, the overall presentation is improved, including limited usage of bullet points.
>
> **Q1: Image Quality in Figure 6**
>
> The CLIP-T metric focuses on prompt alignment rather than visual quality. The quality of images in Figure 6 is attributed SDXL model rather than a deficiency in FairCoT as our method doesn’t alter model weights. For example, SD v1.5 and v2.1 achieved higher-quality images we now include in the appendix. We also included in our report additional comparative images from baseline methods to further demonstrate the capabilities of our approach with detailed image examples (Figures 6,7,8,9 on pages 14-17) .
>
>
> [1] Ruifei He, Chuhui Xue, Haoru Tan, Wenqing Zhang, Yingchen Yu, Song Bai, and Xiaojuan Qi. 2024. Debiasing Text-to-Image Diffusion Models. In Proceedings of the 1st ACM Multimedia Workshop on Multi-modal Misinformation Governance in the Era of Foundation Models (MIS '24). Association for Computing Machinery, New York, NY, USA, 29–36. https://doi.org/10.1145/3689090.3689387
>
> [2]Shrestha, R., Zou, Y., Chen, Q., Li, Z., Xie, Y., & Deng, S. (2024). FairRAG: Fair human generation via fair retrieval augmentation. In Proceedings of the IEEE/CVF Conference on Computer Vision and Pattern Recognition (pp. 11996-12005).
>
> [3] Shen, X., Du, C., Pang, T., Lin, M., Wong, Y., & Kankanhalli, M. (2024). Finetuning text-to-image diffusion models for fairness. In The Twelfth International Conference on Learning Representations.

---

> ### Author Response · Authors · 2024-12-01
> **Follow-Up**
>
> Dear Reviewer wptm, as we are approaching the end of the author-reviewer discussion period, could you please let us know if our responses have adequately addressed your concerns? If there are any remaining comments, we are committed to providing prompt responses before the discussion period concludes. Thank you for your time and consideration.

---

> > ### Comment · Reviewer_wptm · 2024-12-01
> >
> > Thank you for your responses. I still think that the technical contribution of this work is limited. I will keep my current score.

---

### Meta-Review · Area_Chair_VTGs · 2024-12-23

**Metareview:**

This paper presents FairCoT, a framework designed to enhance fairness in text-to-image diffusion models through Chain-of-Thought reasoning within multimodal language models. The authors propose an iterative refinement process incorporating attire-based attribute prediction and bias assessment using entropy scores, claiming improvements in fairness across multiple demographic attributes while maintaining image quality.

All reviewers unanimously gave negative scores. The core technical contribution is modest, primarily combining existing techniques without substantial innovation. The evaluation method raises significant concerns, particularly the circular dependency of using CLIP for both generation and evaluation. The lack of rigorous comparison with established baselines, limited validation beyond automated metrics, and potential reproducibility issues further undermine the paper's scientific contribution. The paper needs a significant revision to address those concerns to reach the publication criteria.

**Additional Comments On Reviewer Discussion:**

See above.

---

### Decision · Program_Chairs · 2025-01-22

Reject